# Induced formation of primordial germ cells from zebrafish blastomeres by germplasm factors

Xiaosi Wang [1,2,3], Junwen Zhu[1,2], Houpeng Wang[1,3], Wenqi Deng[1,2], Shengbo Jiao[1,2], Yaqing Wang[1,2,3], Mudan He[1,2,3], Fenghua Zhang[1,2], Tao Liu[1,2], Yongkang Hao[1,2], Ding Ye[1,2,3] & Yonghua Sun [1,2,3] ✉

The combination of genome editing and primordial germ cell (PGC) transplantation has enormous significance in the study of developmental biology and genetic breeding, despite its low efficiency due to limited number of donor PGCs. Here, we employ a combination of germplasm factors to convert blastoderm cells into induced PGCs (iPGCs) in zebrafish and obtain functional gametes either through iPGC transplantation or via the single blastomere overexpression of germplasm factors. Zebrafish-derived germplasm factors convert blastula cells of *Gobiocypris rarus* into iPGCs, and *Gobiocypris rarus* spermatozoa can be produced by iPGC-transplanted zebrafish. Moreover, the combination of genome knock-in and iPGC transplantation perfectly resolves the contradiction between high knock-in efficiency and early lethality during embryonic stages and greatly improves the efficiency of genome knock-in. Together, we present an efficient method for generating PGCs in a teleost, a technique that will have a strong impact in basic research and aquaculture.

Primordial germ cells (PGCs) are the embryonic progenitor cells of sperm and eggs, which transmit genetic material between generations[1,2]. Therefore, PGCs are ideal target cells for genetic manipulations, such as genome editing, gene knock-in (KI), and transgenesis[3–5]. In several animals, genetically manipulated PGCs have been transplanted into host embryos to produce genetically modified gametes in a technique known as surrogate reproduction by PGC transplantation (PGCT), which is designed to increase the efficiency of genetic manipulations[6,7]. A very limited number of PGCs exist in early embryos across various animals[8–10]. Therefore, how to obtain a large number of PGCs in a given animal is pivotal for efficient surrogate reproduction by PGCT.

There are two different modes of PGC formation in animals, namely preformation and epigenesis[11–13]. Epigenesis, which occurs mainly in mammals, refers to the mechanism through which surrounding tissues send signals to induce specific cell populations to acquire PGC properties. In contrast, preformation is a process of germ cell-autonomous specification that describes the mechanism through which PGCs are specified by maternally inherited germplasm factors. Preformation prevails in fish, amphibians and other oviparous animals[14]. Maternally inherited germplasm factors, such as *vasa*[15], *dazl*[16], *piwil1*[17], *dnd1*[18], *nanos3*[19], and *tdrd*[20,21], are deposited in presumptive primordial germ cells (pPGCs), leading them to segregate from the somatic lineage and eventually form PGCs. At present, the induction of PGC-like cells from embryonic stem cells or pluripotent stem cells has been successfully achieved in mice[22,23], rats[6] and northern white rhinoceros[24], and the induced PGCs (iPGCs) can be further differentiated into functional gametes after transplantation into germ cell-deficient host animals[6,22–24]. However, these induction strategies are all based on the epigenesis model, and there are no successful reports of PGC induction using the preformation model in oviparous species.

Although the zebrafish has emerged as an important animal model for studying vertebrate development[25], human disease[26], and

[1]State Key Laboratory of Freshwater Ecology and Biotechnology, Key Laboratory of Breeding Biotechnology and Sustainable Aquaculture, Institute of Hydrobiology, Innovation Academy for Seed Design, Chinese Academy of Sciences, Wuhan 430072, China. [2]College of Advanced Agricultural Sciences, University of Chinese Academy of Sciences, Beijing 100049, China. [3]Hubei Hongshan Laboratory, Wuhan 430070, China. ✉e-mail: yhsun@ihb.ac.cn

finfish aquaculture[27], it remains challenging to obtain mutants or KI alleles of embryonic lethal genes in zebrafish owning to an obvious trade-off between the embryonic mutation efficiency, the embryonic survival rate, and the germline transmission rate[28]. The transplantation of mutated or KI primordial germ cells into germ cell-depleted host embryos has been shown to be a valuable approach to address this problem[29]. However, this method is usually inefficient, time-consuming, and laborious, and it is difficult to obtain functional gametes using the conventional PGCT approach, mainly due to the limited number of PGCs in donor embryos.

In the present study, a combination of nine germplasm factors (9GMs) was successfully identified from an initial pool of 13 germplasm factors (13GMs) that could efficiently convert blastoderm cells into induced PGCs (iPGCs) in zebrafish. This study obtained iPGC-originated mature spermatozoa either via iPGC transplantation (iPGCT) into a PGC-depleted host or through the overexpression of 9GMs in a single blastomere. Moreover, the combination of genome KI technology and iPGCT perfectly resolves the contradiction between high KI efficiency and embryonic lethality. This is the first time that the preformation strategy has been used to induce PGCs in oviparous animals, which greatly improves the success rate of PGCT and provides an efficient method for PGC induction and directional breeding in other animals using preformation theory.

## Results

### Induction of PGC like cells via 9GMs in vivo

Ectopic PGCs can be induced by germplasm transplantation in Drosophila[12,30], suggesting that a germplasm cocktail may be used for PGC induction in oviparous animals. First, an mRNA cocktail of 13GMs was used, including 9GMs related to PGC specification[14] (*vasa*[15], *dazl*[16], *piwil1*[17], *dnd1*[18], *nanos3*[19], *tdrd6*[20], *tdrd7a*[21], *dazap2*[31] and *buc*[32]) and four germplasm factors (4GMs) related to PGC migration (*rgs14a*[33], *cxcr4a*[34], *cxcr4b*[35,36] and *ca15b*[33,37]) for the PGC induction experiment in zebrafish (Fig. 1a). GFP-UTR*nanos3*, which is specifically expressed in endogenous PGCs (ePGCs)[28,38], was used to visualize putative PGCs. After the injection of 13GMs into one-cell-stage zygotes, almost all blastula cells became GFP-positive at 9 hours post-fertilization (hpf), whereas only a small number of cells in control or *buc*-overexpressed embryos were GFP-positive (Fig. 1b). Interestingly, when the 4GMs related to PGC migration were removed, the remaining 9GMs were still able to induce putative PGCs efficiently. To induce PGCs more efficiently, different concentrations of germplasm mRNA were injected, and it was found that GFP-UTR*nanos3* was weakly positive at low doses (25 pg mRNA per factor), while the proliferation of embryonic cells was strongly inhibited at high doses (100 pg mRNA per factor). In contrast, iPGC could be effectively induced only at moderate doses (50 pg mRNA per factor) (Supplementary Fig. 1a). Therefore, moderate doses of mRNA were used for iPGC induction in the subsequent experiments.

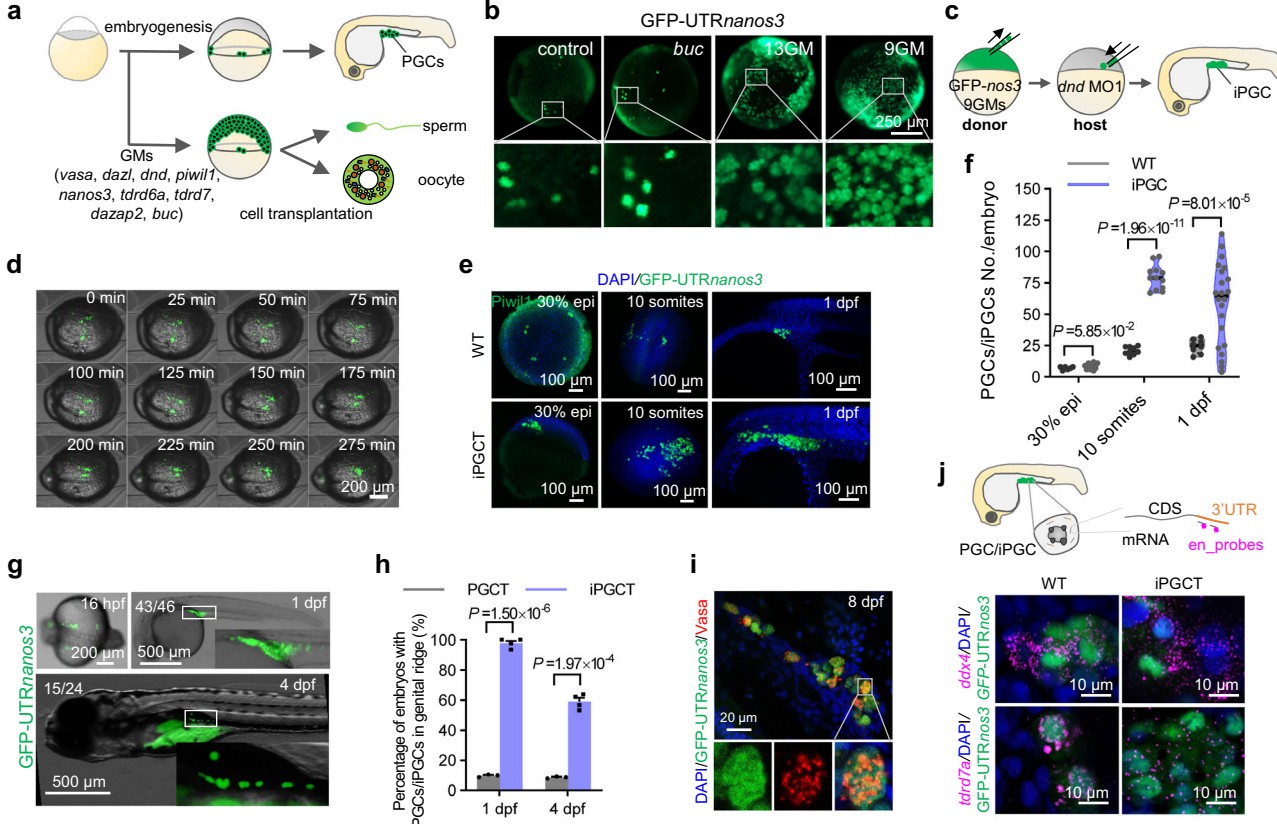

**Fig. 1 | Generation of iPGC via germplasm in vivo. a** Scheme for the iPGC induction in vivo using germplasm. **b** At the 1-cell stage, zebrafish embryos were injected with *buc* (200 pg), 13GM (50 pg per fraction), or 9GM (50 pg per fraction) mRNA to induce iPGCs. GFP-UTR*nanos3* was used to label putative PGCs. The embryos were photographed at 90% epiboly stage. **c** Schematic diagram for iPGCT. **d** Time-lapse of iPGC migration to the genital ridges after iPGCT. GFP-UTR*nanos3* was used to label iPGCs. **e** GFP-UTR*nanos3* was used to label PGCs and iPGCs, except that immunofluorescence against Piwil1 was used to visualize endogenous PGCs of wild-type embryos at 30% epiboly. **f** Number of PGCs in wild-type and iPGCT embryos at different stages. Each point represents an independent sample

($n \geq 6$). **g** iPGCs migrated to the genital ridge of the host embryo. **h** Success rates of iPGCT and conventional PGCT at 1 dpf and 4 dpf. Each point represents an independent experiment ($n \geq 3$). **i** Immunofluorescence detection of Vasa protein in GFP-UTR*nanos3*-positive iPGCs of iPGCT embryos at 8 dpf. **j** Probes were designed at the UTR of mRNA (en_probes) to distinguish neonatal mRNA, and single molecule in situ hybridization was used to detect the neonatal germplasm mRNA of *ddx4* and *tdrd7a* in the GFP-UTR*nanos3*-positive iPGCs and ePGCs at 1 dpf. A representative example of three replicate is shown. All data are presented as mean values ± SEM. Two-tailed Student's *t*-test was used to calculate the *P* values.

In zebrafish, *Drosophila* and other animals, PGCs are specified in the embryo and must migrate to the site of gonadal development for proliferation and differentiation[37,39]. Because the embryos injected with the mRNA cocktail could not survive beyond 12 hpf, the presumptive iPGCs at 4 hpf were transplanted into the 4 hpf embryos in which ePGCs had been depleted by injection of *dnd* morpholino1 (*dnd* MO1) (Fig. 1c). The iPGCs induced by 13GMs or 9GMs rapidly migrate to the genital ridges of the host during embryonic development (Fig. 1d, Supplementary Fig. 1b and Supplementary Movie 1). There was no significant difference in the number of iPGCs induced by nine and 13 factors (Fig. 1b and Supplementary Fig. 1b). This further suggests that the 4GMs related to PGC migration were not involved in the key germplasms for PGC induction. Therefore, 9GMs were identified for PGC induction in the following research. Compared with ePGCs, iPGCs exhibited a faster proliferation rate, resulting in a large number of iPGCs in iPGCT embryos at the 10-somites stage and 1-day post-fertilization (dpf) (Fig. 1e, f, Supplementary Fig. 1c and Supplementary Movie 2). Therefore, it was only necessary to transplant about 10 cells in the iPGCT experiment, which could still greatly improve the efficiency and success rate of PGC transplantation (Fig. 1g, h and Supplementary Movie 3). In contrast, with conventional PGCT, because it was impossible to accurately aspirate the donor PGCs, more donor cells (50–100 cells) were generally aspirated, and the success rate of transplantation through PGCT was only about 10% (Fig. 1h)[29]. In addition, the number of iPGCs in the iPGCT embryos at 1 dpf varied widely, and the embryos with a low number of iPGCs tended to develop to PGC-less embryos at 4 dpf, which finally became infertile adults (Fig. 1f, h and Table 1), suggesting the presence of iPGCs in embryos at 4 dpf is vital for the success of iPGCT.

To further demonstrate that iPGCs share the same biological characteristics as ePGCs, iPGCs were transplanted into *Tg(piwil1:egfp-UTRnanos3)* embryos, which could label ePGCs[38]. iPGCs (red) and ePGCs (green) migrated together to the genital ridge (Supplementary Fig. 1d and Supplementary Movie 4). Vasa, an important germ granule-associated protein[40], was dispersed around the nucleus of GFP-expressed iPGCs at shield stage and aggregated into granules in iPGCs at 8 dpf (Fig. 1i; Supplementary Fig. 1e), while the host embryos injected with *dnd* MO did not contain any endogenous PGC at 1 dpf and 8 dpf (Supplementary Fig. 1f, g), further confirming that the germ cells in the host embryos were derived from the transplanted iPGCs. Next, we wondered whether and when the iPGCs initiate the expression of germplasm genes. *ddx4* and *tdrd7a* are important germplasm genes in PGC[21,41] that were detected using single molecule in situ hybridization (smFISH) technology[42]. To distinguish the endogenously transcribed mRNA (neonatal mRNA containing endogenous 3′UTR) from the exogenously loaded mRNA (without endogenous 3′UTR), probes were designed in the UTR regions of mRNA (en_probes) and CDS (ex_probes), respectively (Supplementary Fig. 1h, i). The ex_probes signals of *tdrd7a* and *ddx4* were detected from the 30% epiboly to 5-somites stage, while en_probes signals were detected only at 5-somites and 1

dpf stage, and almost all iPGCs could activate endogenous PGC genes (Fig. 1j and Supplementary Fig. 1h, i). These data suggest that iPGCs initiate the expression of germplasm genes at 5-somites stage, and a majority of 9GM-induced iPGCs have PGC properties after transplantation into the host embryos.

## Functional gametes generated by iPGC transplantation

To monitor the lifelong development of iPGCs after transplantation, *Tg(cmv:GFP)* and *Tg(cmv:mCherry)* embryos were employed as donors and hosts, respectively. After transplanting *Tg(cmv:GFP)* iPGCs into PGC-depleted *Tg(cmv:mCherry)* embryos, chimeric larvae expressing GFP in germ cells and mCherry in somatic cells were successfully generated (Fig. 2a, b). During the development of iPGCT embryo, iPGCs proliferated and differentiated within the host embryos (Supplementary Fig. 2a–d). At 32 dpf, the GFP-positive germ cells in the iPGCT gonads also exhibited a high level of Vasa expression (Fig. 2c), indicating the survival and differentiation of GFP-positive iPGCs into germ cells in the host fish at 32 dpf. Upon reaching adulthood, all iPGCT fish developed into fertile males, displaying testes of unequal size on both sides of their bodies (Supplementary Fig. 2e). This asymmetry could be attributed to the varying locations of the implanted iPGCs within the host embryos (Supplementary Fig. 1d), resulting in the unequal migration of iPGCs toward two sides of the embryo, i.e., a high number of iPGCs on one side and few or no iPGCs on the other side. However, this disparity did not affect the gonad structure or germ cell development. As shown in Fig. 2d, the iPGCT gonads contained spermatogenic cells at various stages, such as spermatocytes (SCs) and spermatids (SZs). Similar to control gonads, germ cells within iPGCT gonads underwent both meiosis and mitosis (Fig. 2e). Moreover, the expression levels of germ cell-specific genes, gonadal somatic cell-specific genes, meiosis-related genes, and mitosis-related genes did not significantly differ between the control and iPGCT testes (Supplementary Fig. 2f). Additionally, iPGCT fish produced mature spermatozoa with head diameters and tail lengths comparable to those of the controls (Supplementary Fig. 2g–i). The sperm volume of iPGCT fish was about half that of wild-type fish (Supplementary Fig. 2j), which was potentially attributed to a substantial number of iPGC-derived germ cells existing on only one side of the testis. Interestingly, germ cells in iPGCT gonads were GFP-positive, while gonadal somatic cells were mCherry-positive (Fig. 2f), suggesting that the transplanted *Tg(cmv:GFP)* iPGCs eventually differentiated into spermatozoa but not somatic cells. Finally, a chimera fish was obtained that expressed mCherry throughout the body but produced GFP-expressing spermatozoa (Fig. 2g). When this chimera mated with a wild-type female, they produced GFP-expressing progeny. Notably, iPGCT allowed for a 100% gamete retrieval rate from donors (Table 1), greatly improving the efficiency of PGCT-mediated surrogate production. The F2 progeny resulting from iPGCT fish still developed normally and expressed GFP throughout the body (Fig. 2g). These results demonstrate that almost all embryonic cells can be induced into iPGCs by 9GMs, and that donor-derived gametes can be efficiently obtained using iPGCT.

## Functional gametes produced by single-blastomere-induced iPGCs

To further demonstrate that 9GMs can convert any embryonic cell into iPGC, the localized injection of 9GMs into a single blastomere was performed at the 128-cell stage. At the 16-cell stage, the ectopic injection of *buc* mRNA in a clone was able to recruit the germplasm already present in the cleavage furrow, inducing it into PGCs[32]. However, *buc* mRNA alone failed to induce PGCs at the 128-cell stage (Fig. 3a, b), suggesting the absence of germplasm factors near this clone at this stage. In contrast, the localized injection of 9GMs into one blastomere efficiently induced iPGCs, which then correctly migrated to the genital ridge (Fig. 3a, b), suggesting that 9GMs injection likely led to expression of the PGC motility module. Importantly, the induction

## Table 1 | Efficiency of iPGCT and PGCT

| Transplantation strategies | | PGC-positive embryos/survived embryos at 1 dpf (%) | PGC-positive embryos/survived embryos at 4 dpf (%) | Positive fertile adults/adults originated from PGC-positive embryos at 4 dpf (%) |
|---|---|---|---|---|
| *Dr* | PGCT | 13/128 (10.16%) | 6/65 (9.23%) | 4/4 (100%) |
| | iPGCT | 43/46 (93.48%) | 23/37 (62.16%) | 15/15 (100%) |
| | | 42/43 (97.67%) | 23/36 (63.89%) | 19/19 (100%) |
| | | 31/31 (100%) | 17/30 (56.67%) | 14/14 (100%) |
| *Gr* | PGCT | 2/91 (2.20%) | 1/83 (1.20%) | 1/1 (100%) |
| | iPGCT | 132/135 (97.78%) | 38/65 (58.46%) | 3/16 (18.75%) |
| | | 120/120 (100%) | 49/77 (63.64%) | 4/22 (18.18%) |

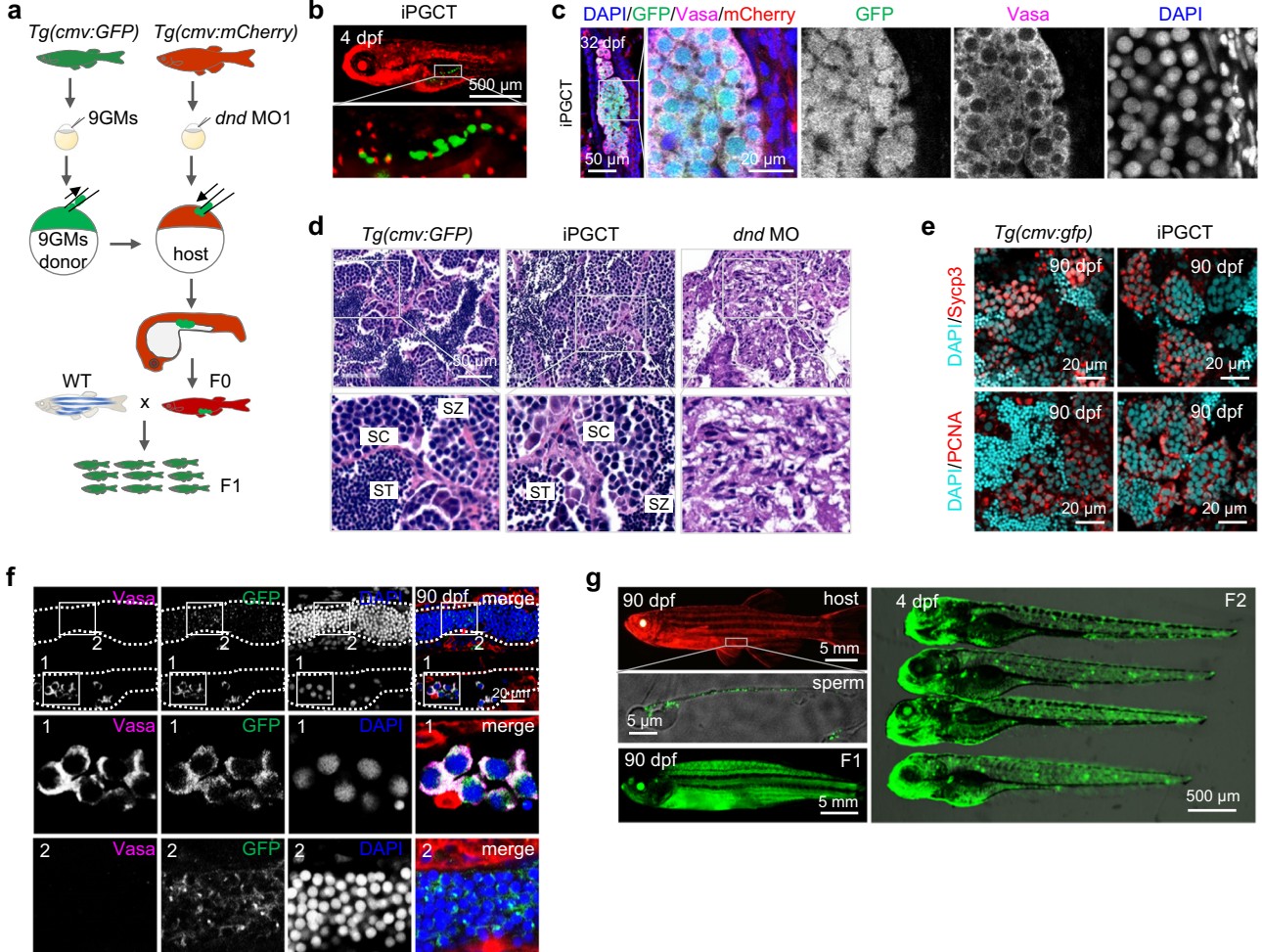

**Fig. 2 | Functional gametes were obtained using iPGCT. a** Schematic diagram for iPGCT using *Tg(cmv:GFP)* embryos as iPGCT donors and *Tg(cmv:mCherry)* embryos as iPGCT hosts. **b** After iPGC transplantation, host embryos expressed mCherry and germ cells expressed GFP. **c** Immunofluorescence detection of Vasa in GFP-positive germ cells of iPGCT gonads at 32 dpf. Note that gonadal somatic cells expressed mCherry. **d** H&E staining of *Tg(cmv:GFP)* and iPGCT testes. SC spermatocyte, ST spermatid, SZ spermatozoa. **e** Immunofluorescence detection of Sycp3 and PCNA showing normal meiosis and mitosis of germ cells in iPGCT testis and control testis at 90 dpf. **f** Immunofluorescence detection of Vasa in GFP-positive germ cells of iPGCT testes at 90 dpf. Rectangular boxes 1 and 2 show the development of gonads on both sides of the iPGCT embryo, respectively. **g** The iPGCT fish expressing mCherry produced sperm expressing GFP, and F1 and F2 generations expressing GFP were obtained. A representative example of three replicate is shown.

of iPGCs was not confined by location and could occur in any cell, whether situated at the animal pole or the margin of the embryos (Fig. 3a, b and Table 2). These results suggest that 9GMs are sufficient to induce PGCs in any cell at 128-cell stage. To further confirm the PGC properties of these induced cells, CRISPR/Cas9-mediated genome editing technology was employed to genetically label these cells (Supplementary Fig. 1a). If gene-edited gametes were obtained, it would confirm the transformation of 9GM-induced cells into germ cells. For this purpose, three representative genes were selected for editing, namely *tyrosinase* (*tyr*), a gene linked to pigment development that features an easily observable mutant phenotype, and *bmp7a* and *pou5f3*, which are pivotal genes in early embryonic development, with mutants that display severe developmental defects[43]. As shown in Fig. 3c, d and Supplementary Movie 5, most of the blastomeres injected with 9GMs divided quickly and migrated correctly to the genital ridges, suggesting that the blastomeres were induced into iPGCs by 9GMs. Finally, one iPGC-positive embryo co-injected with *tyr* gRNA, two with *bmp7a* gRNA and one with *pou5f3* gRNA survived to adulthood. When these fish were crossed with wild-type fish, all of them were able to produce genome-edited heterozygous embryos with an efficiency between 10% and 20% (Supplementary Fig. 3b, c). In contrast, when localized injection was performed without iPGC induction, the GFP-positive cells distributed throughout the body except for the genital ridge. As a result, no offspring with mutations were obtained (Fig. 3e, f). Mutants obtained using these methods exhibited the same developmental defects (Supplementary Fig. 3d–f) as previously reported[43,44].

To increase the efficiency of 9GM induced PGCs by single-blastomere injection, a low dose (20 μM) of *dnd* MO2[45] was injected into one-cell stage embryos to eliminate endogenous PGCs before the single-blastomere injection of 9GMs, *Cas9* mRNA, and gRNAs against *pou5f3* or *tyr*. The *dnd* MO2 only interferes with the translation of endogenous *dnd* mRNA but does not affect the translation of exogenous *dnd* mRNA (Fig. 3g). By eliminating the endogenous PGCs before the single-blastomere induction of iPGCs, the mutation efficiency of the F0 gametes was significantly improved to over 80% (Fig. 3h). Overall, these results demonstrate that 9GMs are sufficient to induce any blastomere into iPGCs in vivo.

## Xenogametes generated by iPGCT
Having confirmed the ability of 9GMs to induce iPGCs in zebrafish embryos, this study then aimed to investigate whether the underlying

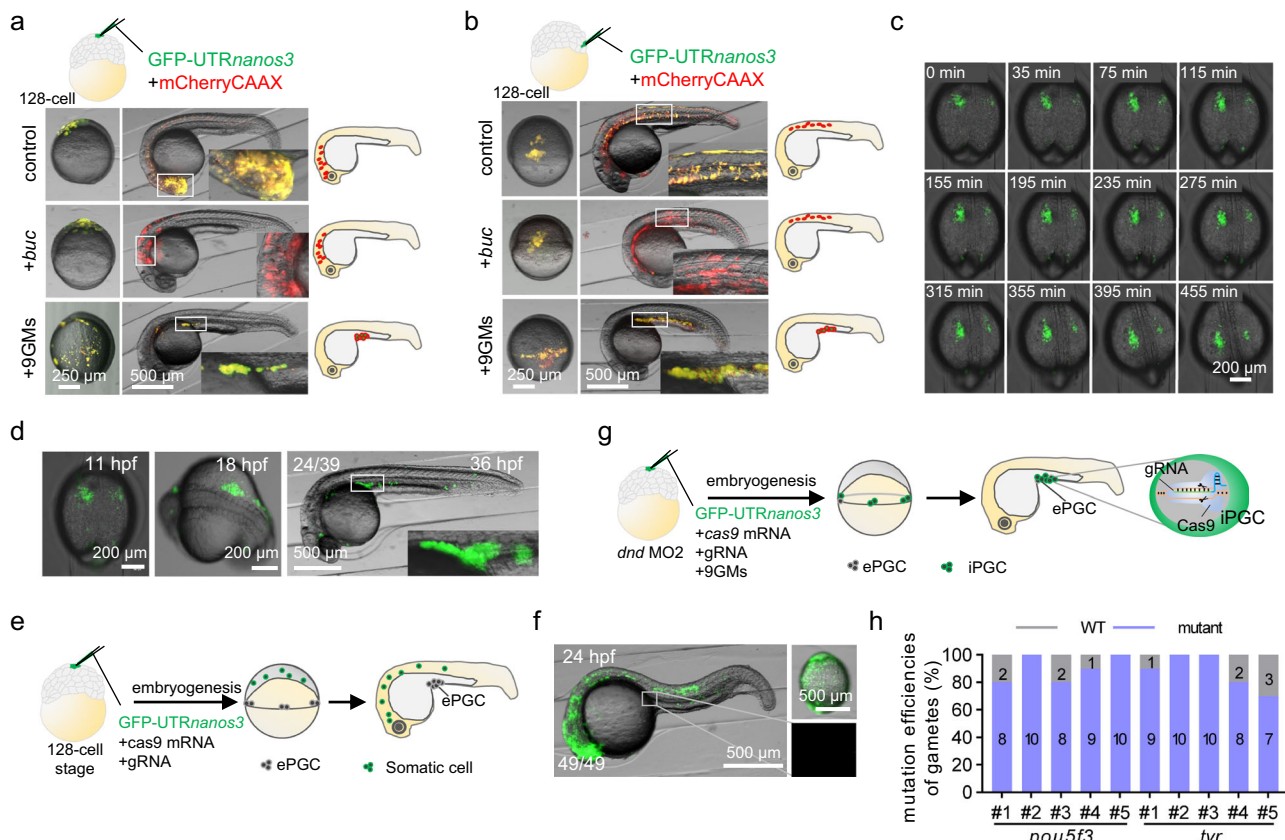

**Fig. 3 | Functional gametes were produced by the single blastomere induction of iPGCs.** Injection of 9GM or *buc* mRNA into a single blastomere at the animal pole (**a**) or margin (**b**) of a 128-cell embryo. GFP-UTR*nanos3* was used to label iPGCs, and mCherryCAAX was used to label injected cells. **c** Time-lapse of iPGC migration to the genital ridges after single blastomere injection. **d** Single blastomere-induced iPGCs migrated to the genital ridge. **e, f** Genome editing was performed in one blastomere at 128-cell stage without iPGC induction. Eventually, these genome-edited cells migrated to other parts of the body besides the genital ridge. **g** Schematic representation of simultaneous genome editing and iPGC induction in

a single blastomere at 128-cell stage. Embryos were injected with a low dose of *dnd* MO2 (20 μM) at 1-cell stage to eliminate the endogenous PGC, and the single blastomere induced iPGCs developed into genome-edited gametes. **h** Mutation efficiencies of gametes originated from single blastomere induced iPGCs. For each fish, a total of 10 embryos obtained from the hybridization of genome-edited sperm originated from single blastomere induced iPGCs and wild-type eggs were analyzed. The mutation efficiency of 5 F0 fish (#1–#5) gametes is shown for each gene. A representative example of three replicate is shown.

mechanism for 9GM-induced iPGCs was conserved in other fish species. *Gobiocypris rarus* (*Gr, gobioninae*), a small cyprinid fish endemic to China, has been used for cross-subfamily germ cell transplantation with zebrafish (*Danio rerio, Dr*)[46]. Here, zebrafish-derived 9GMs were used to induce iPGCs in *Gr*, and *Gr*-derived iPGCs (*Gr*_iPGCs) were transplanted into zebrafish embryos, i.e., zebrafish was used to produce gametes of *Gr* (*Gr*_iPGCT_*Dr*) (Fig. 4a). As in zebrafish embryos, almost all embryonic cells can be induced as GFP-UTR*nanos3* positive cells (Fig. 4b). When *Gr*_iPGCs were transplanted into ePGC-depleted zebrafish embryos (*Gr*_iPGCT_*Dr*), these *Gr*_iPGCs were able to migrate correctly to the zebrafish genital ridges at a rate of almost 100%. The percentage of embryos with PGCs in the genital ridge using iPGCT was much higher than that using traditional PGCT (Fig. 4c–e and Supplementary Movie 6). In addition, the number of positive cells that migrated to the genital ridge in *Gr*_iPGCT_*Dr* was much higher than that in *Gr*_PGCT_*Dr* and WT (Fig. 4d). However, the number of GFP-positive cells gradually decreased during the development of the host zebrafish (Fig. 4f). Immunofluorescence showed that the germ cells in *Gr*_iPGCT_*Dr* were divided into two types: one type (5/6) expressed Vasa weakly, the other type (1/6) expressed Vasa as strongly as *Gr*, and more than 10 germ cells were grouped together (Fig. 4g). The success rate of iPGC transplantation (61.05% on average) was much higher than that of the conventional PGC transplantation (1.2%) (Table 1). It is noteworthy that only about one-sixth of the adults that originated from PGC-positive 4 dpf embryos produced gametes, which was

consistent with the fact that only one-sixth of embryos were strong positive vasa antibodies in germ cells at 60 dpf. This may have occurred because germplasm factors derived from zebrafish were utilized, leading to the insufficient activation of *Gr* germplasm-related genes in xenografts. The sperm morphology of *Gr*_PGCT_*Dr* was also more similar to that of *Gr* (Fig. 4h and Supplementary Fig. 4a, b). Finally, zebrafish were used to rapidly produce *Gr* sperm via cross-subfamily iPGCT and obtain *Gr* F1 progeny, which grew up to adulthood and produced F2 offspring normally (Fig. 4i and Supplementary Fig. 4c–e). Given the efficient iPGC induction across subfamily species, it is reasonable to assume that the in vivo induction of iPGCs using 9GMs is highly conserved in teleost fish.

## iPGCT greatly improved the KI efficiency of gametes

CRISPR/Cas9-mediated genome knockout and KI technologies enable the rapid and directed modification of biological traits[47,48]. However, the trade-offs between high KI efficiency and high embryo survival, along with the low germline transmission rate of KI alleles, necessitate extensive screening efforts to obtain KI progeny. This challenge substantially constrains the applicability of KI in fish. In theory, through iPGC induction and transplantation technology, somatic cells with high KI efficiency could be converted to germ cells, thereby greatly improving the efficiency of KI gamete transmission. Here, genome KI was performed on the genomic loci of *mpx*[49], *sox19b*, and *nanog*[50,51] using microhomology-mediated end-joining (MMEJ, 6 bp

**Table 2 | Proportion of cells in different positions of the embryo induced to PGC**

| Injection location | Sample | Location of positive cells | | |
|---|---|---|---|---|
| | | Genital ridge (%) | Trunk (%) | Brain (%) |
| Animal pole | – | 0 | 0 | 100 ($n$ = 44) |
| | +*buc* | 0 | 0 | 100 ($n$ = 38) |
| | +9GM | 100 ($n$ = 48) | 0 | 10.42 ($n$ = 5)[a] |
| Margin | – | 0 | 100 ($n$ = 58) | 0 |
| | +*buc* | 0 | 100 ($n$ = 66) | 0 |
| | +9GM | 100 ($n$ = 62) | 12.9 ($n$ = 11)[a] | 0 |

Different combinations of mRNA were injected at the animal pole of 128-cell stage embryos. mCherryCAAX was used to label injected cells, GFP-UTR*nanos3* was used to label iPGCs, and *buc* or 9 germplasm (9GM) was used to induce iPGCs.
[a]A small number of positive cells migrated to the head or trunk, and most of the positive cells migrated to the genital ridge.

microhomology) or non-homologous DNA end joining (NHEJ) approaches[52] (Fig. 5a). Although the injection of a high dose of KI reagents could considerably increase the KI efficiencies in F0 embryos, this generally led to serious developmental defects (Fig. 5a–d; Supplementary Fig. 5a–c). The combination of KI and iPGCT offers a perfect solution to this challenge of early mortality in high-efficiency KI embryos. As shown in Fig. 5e, the KI efficiency of the donor should be maximized, regardless of embryonic survival and the expression of fluorescent proteins, and surrogate fish could be harnessed to effectively generate gametes harboring KI alleles (Fig. 5e, f, Table 3). In accordance with this, the combination of KI and iPGCT substantially improves the efficacy of the germline transmission of KI alleles, achieving a rate of over 90% (Fig. 5g–i, Supplementary Fig. 5d–j and Table 3). In conclusion, the combination of KI and iPGCT technologies greatly improves the efficiency of the genome KI and germline transmission of zebrafish. Regardless of the survival of the donor embryos, the concentration of KI components can be considerably increased to reach as high a KI efficiency as possible. Moreover, if the KI efficiency is

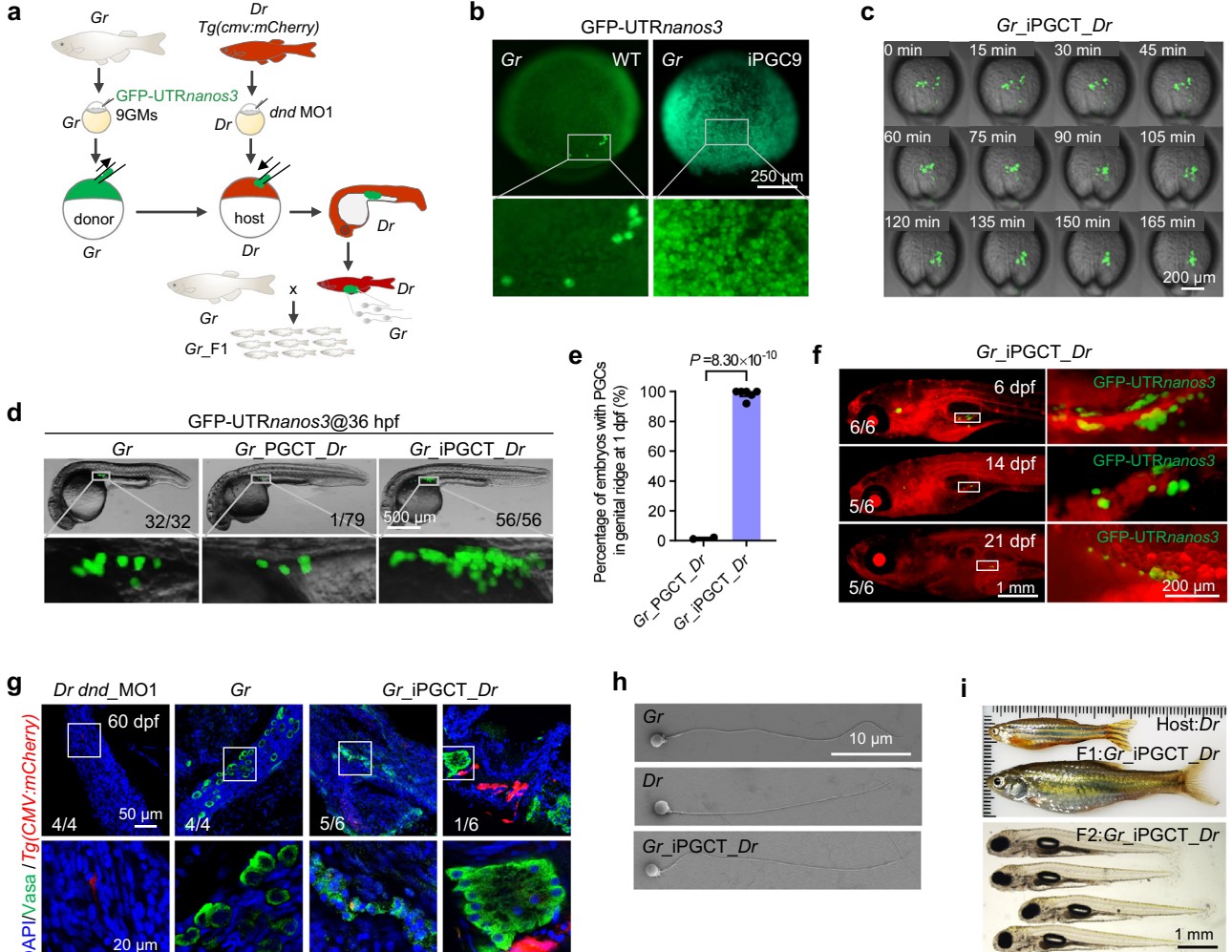

**Fig. 4 | Generation of xenogametes by iPGCT. a** Schematic diagram for iPGCT using *Gobiocypris rarus* (*Gr*) as the donor and *Tg(cmv:mCherry)* zebrafish (*Danio rerio*, *Dr*) as the host. **b** At the 1-cell stage, *Gr* embryos were injected with zebrafish-derived 9GMs (50 pg per mRNA) to induce iPGCs. GFP-UTR*nanos3* was used to label iPGCs. The embryos were photographed at 90% epiboly. **c** Time-lapse of *Gr*_iPGCs migration to the genital ridges of *Dr* after transplantation. **d** Fluorescent image of conventional PGCT embryos and iPGCT embryos at 36 hpf. *Gr*_PGCT_*Dr*, *Gr* PGCs transplanted to PGC-depleted *Dr* embryos; *Gr*_iPGCT_*Dr*, 9GMs induced *Gr* iPGCs transplanted to PGC-depleted *Dr* embryos. **e** Success rate of iPGCT and conventional PGCT at 1 dpf. Each point represents an independent experiment ($n ≥ 2$), and at least 77 embryos were manipulated per experiment. A representative example of three replicate is shown. All data are presented as mean values ± SEM. Two-tailed Student's *t*-test was used to calculate the *P* values. **f** Fluorescence imaging of GFP-UTR*nanos3* positive *Gr*-derived iPGCs (*Gr*_iPGCs) in *Tg(cmv:mCherry)* zebrafish host. **g** Immunofluorescence detection of Vasa-positive germ cells in *dnd*_MO1 injected *Dr* (*Dr dnd*_MO1), *Gr*, and *Gr*_iPGCT_*Dr* testes at 60 dpf. **h** Morphology of the sperm of *Gr*, *Dr* and *Gr*_iPGCT_*Dr*. **i** *Gr*_iPGCT_*Dr* sperm generated by *Dr* hosts produced F1 and F2 generations of *Gr* normally.

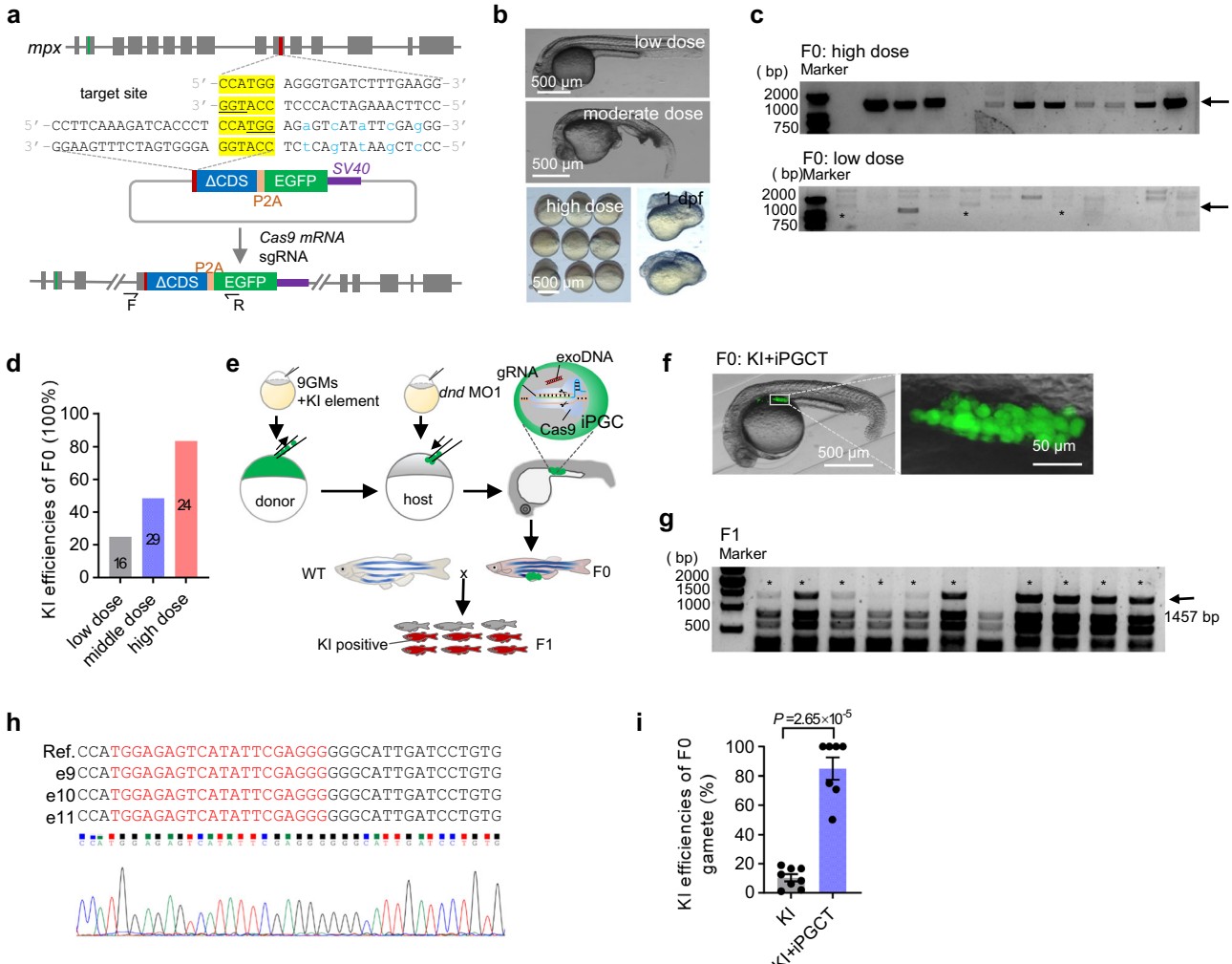

**Fig. 5 | iPGCT greatly improved the knock-in (KI) efficiency of gametes.**
**a** Schematic diagram of KI strategy mediated by microhomology-mediated end-joining (MMEJ). GFP was inserted into exon 10 of *mpx* and the missing CDS sequence was added in the KI vector. The underlined bases are PAM regions, the yellow bases are microhomology sequences, and the blue bases are bases with synonymous mutations. **b** With the increase of KI reagent doses, the embryos displayed increasingly serious abnormalities. Low dose (50 pg gRNA, 50 pg plasmid, and 500 pg *cas9* mRNA); moderate dose (100 pg gRNA, 100 pg plasmid, and 1000 pg *cas9* mRNA); and high dose (200 pg gRNA, 200 pg plasmid, and 1000 pg *cas9* mRNA). **c** Single embryos were used to evaluate the efficiency of KI events. The arrow indicates the positive bands (1457 bp), and the asterisk represents the positive embryos at low dose. **d** KI efficiency of F0 embryos at different doses. The

number on the column represents the total number of embryos tested. **e** Schematic diagram showing KI combined with iPGCT to efficiently produce KI progeny. **f** The iPGCT embryos harboring KI iPGCs were obtained according to the process illustrated in **e**. **g** F0 was mated with the wild-type, and 12 embryos were selected for identification one by one to evaluate the KI efficiency of F0 gamete. The arrow indicates the positive bands (1457 bp), and the asterisk represents the positive embryo. **h** Sequencing results of positive bands. gRNA target locations are marked in red. **i** KI efficiencies of F0 gametes producing positive F1. Each dot represents the rate of KI gametes from one F0 fish ($n \geq 7$). A representative example of three replicate is shown. All data are presented as mean values ± SEM. Two-tailed Student's *t*-test was used to calculate the *P* values.

high enough, there is no need to screen positive embryos based on fluorescence in the F0 generation, which favors more complex KI design and suitability of low-expression genes KI.

## Discussion

The combination of genome editing and surrogate broodstock technologies has proven to be an innovative technology for rapid and targeted fish breeding[27,53,54]. However, there are two problems that need to be solved: the limited success rate of surrogate production and the challenge of the efficient editing of donor genomes. Germline stem cell transplantation (GSCT) has been used to obtain intra-species or cross-species gametes, but it is difficult to obtain genome-edited gametes[46,55–57]. While the donor cells of PGCT are derived from early embryos, in which genome editing can be readily performed, the low number of donor PGCs remains a significant obstacle for successful

PGCT-based surrogate production[28,58]. In this study, an iPGC approach to successfully convert any blastomere into iPGCs was developed, thereby greatly improving the efficiency of iPGCT. This approach may create an efficient method for germline-targeted genome editing and surrogate breeding in zebrafish and other farmed fish species.

Preformation, a process of specifying PGC through maternally inherited germplasm factors, constitutes a cell-autonomous behavior[11,13]. In zebrafish and medaka fish, previous studies have focused on PGC induction via single germplasm factors and failed in PGC induction from the mechanism of PGC preformation[32,59]. However, the PGC induction technology in this study directly supplies the germplasm factors necessary for PGC specification, in contrast to Buc overexpression, which induces a small number ectopic PGCs through recruiting existing germplasm factors[32]. This is why Buc cannot induce PGCs at the 128-cell stage, whereas 9GMs can (Fig. 3a, b). In addition,

**Table 3 | Positive rate of offspring with different strategies**

| Gene | Method | PGC-positive embryos/manipulated embryos (4 dpf) | Fluorescence positive embryos/injected embryos | Caudal fin positive adults (KI)/tested adults | Fertile adults (KI + iPGCT)/adults originated from PGC-positive embryos at 4 dpf | F0 adults producing positive F1/tested adults (%) | F0 adults producing positive F1/manipulated embryos (%) |
|---|---|---|---|---|---|---|---|
| *sox19b* | KI | – | 39/403 | 4/39 | – | 2/39 (5.13%) | 2/403 (0.50%) |
| | KI +iPGCT | 20/35 | – | – | 8/19 | 4/19 (21.05%) | 4/35 (11.43%) |
| *mpx* | KI | – | – | 14/384 | – | 8/384 (2.08%) | 8/384 (2.08%) |
| | KI +iPGCT | 15/23 | – | – | 7/12 | 7/12 (58.33%) | 7/23 (30.43%) |
| *nanog* | KI | – | 37/339 | 4/37 | – | 1/37 (2.70%) | 1/339 (0.29%) |
| | KI +iPGCT | 20/32 | – | – | 15/17 | 1/17 (5.88%) | 1/32 (3.13%) |

due to the small number of endogenous PGCs (about 8 cells) at 30% epiboly stage (Fig. 1e), it is even impossible to accurately aspirate the PGCs from donor embryos for transplantation into 2–3 host embryos. Therefore, 50–100 cells are typically required for conventional PGCT to increase the chance of obtaining donor-derived PGCs[29]. Unlike conventional PGCT, almost all blastoderm cells could be induced into iPGCs by 9GMs and the iPGCs even exhibited an accelerated proliferation rate. Thus, only 10–20 donor cells are required for iPGCT experiments. Compared with conventional PGCT (1 donor to 1–2 hosts), iPGCT enables the transplantation of iPGCs from 1 donor embryo to at least 30 host embryos (Supplementary Movie 3), which greatly improves the efficiency of transplantation.

It is worth noting that all iPGCT embryos developed into males in this study. It is known that zebrafish female differentiation is promoted by a sufficient number of PGCs[38], whereas a lack of PGCs leads to male development[60]. In the present study, although all iPGCs initiated germplasm mRNA expression during somitogenesis (Fig. 1j and Supplementary Fig. 1h, i), the number of iPGCT germ cells decreased markedly from 1 dpf to 4 dpf (Fig. 1f, h), and declined continuously from 4 dpf to 12 dpf (Supplementary Fig. 2a–d). After 12 dpf, the number of germ cells in iPGCT embryos was generally lower than that in the control embryos (Supplementary Fig. 2a–d). Therefore, it is likely that the iPGCT embryos did not contain sufficient germ cells to support female development, although in some cases the germ cells could develop into stage I oocytes (Supplementary Fig. 2d). In xenografted chimeric gonads of *Gr*_iPGCT_*Dr*, immune rejection might be one of the main reasons for the low success rate, because immune homeostasis has been shown to play an essential role in regulating gonadal development and gametogenesis[61]. In addition, the strategy developed in the present work utilized the 3′UTR of sv40, potentially resulting in the excessive stabilization of exogenous 9GMs mRNA over an extended period. This prolonged stabilization of 9GM mRNA might hinder the normal development of germ cells from 1 dpf to 25 dpf, which should be accompanied by a dynamic change of mRNAs encoding germplasm factors[16,62,63].

In future studies, the 9GMs should be removed individually to identify a combination of key germplasm factors for iPGC induction in zebrafish, just like the strategy employed in screening key factors for induced pluripotent stem (iPS) cells[64]. In cross-species iPGCT experiments, given the sequence differences between different species, donor species-derived germplasm factors should be optimized for high-efficiency PGC induction. To address the potential immune rejection of transplanted iPGCs, immunodeficient or immunotolerant hosts could offer a promising avenue for achieving high-efficiency iPGC transplantation[65].

## Methods

### Ethics statement
All animal experiments were conducted according to the standard animal guidelines approved by the Animal Care Committee of the University of Chinese Academy of Sciences and the Institute of Hydrobiology, Chinese Academy of Sciences.

### Zebrafish maintenance
Both zebrafish and *Gobiocypris rarus* used in this study were raised at the China zebrafish Resource Center (CZRC) and maintained under a 14-h light/10-h dark cycle at 28.5 °C. Our experiments were carried out using 3-month-old zebrafish from wild-type stocks with AB genetic background and transgenic line *Tg(cmv:GFP)* and *Tg(cmv:mCherry)*. iPGC induction was performed in zebrafish embryos at 1-cell stage, and iPGC transplantation was performed when the embryos developed to sphere stage.

### iPGC induction and iPGC transplantation
As shown in Fig. 1a, c, nine zebrafish-derived germplasm factors and GFP-UTR*nanos3* mRNAs were precisely injected into zebrafish cells at 1-cell stage, which is essential for efficient iPGC induction. At 4 hpf, embryos whose entire blastula was GFP-positive were used as donors. The hosts were contemporaneous embryos injected with *dnd* morpholino1 (5′-GCTGGGCATCCATGTCTCCGACCAT-3′, 50 μM)[28]. iPGCT followed the same procedure as previously published cell transplantation methods[66]. Briefly, more than 500 cells were aspirated and transplanted to the animal pole edge of the hosts, with 10-20 cells per host. One donor embryo-derived iPGCs were usually transplanted into at least 30 host embryos (Supplementary Movie 3). The conventional PGC transplantation procedure was as described previously[28,29,67], and 50–100 cells from the margin of the donor embryo were transplanted to the margin of a similarly staged host.

### RNA synthesis
The CDSs of nine germplasm factors (9GM: *ddx4* (NM_131057), *dazl* (NM_131524), *dnd1* (NM_212795), *piwil1* (NM_183338), *nanos3* (NM_131878), *tdrd6* (XM_688932), *tdrd7a* (NM_001099343), *dazap2* (NM_199793), and *buc* (NM_001256780)) were inserted into the pCS2+ vector. Next, large-scale mRNA was generated using the SP6 Kit (Invitrogen). gRNA used in this study was generated using T7 Kit (Thermo) and purified using LiCl. *bmp7a-gRNA* target:

AGACTGAATGTCATTATCCA; *tyr*-gRNA target: GGACTGGAG-GACTTCTGGGG; *pou5f3*-target: GGGTGAACTACTACACGCCA[43].

### Sperm preparation for scanning electron microscope (SEM)

The procedure used to prepare sperm for SEM was described in the previously published article[46]. Briefly, samples fixed with 2.5% glutaraldehyde overnight were washed with phosphate buffered saline (PBS) and dropped onto cell slides. After the samples were naturally dried, the samples were dehydrated in a gradient with 10%, 30%, 50%, 70%, 80%, 90%, 95%, 100% and 100% ethanol. The dried samples were sprayed with gold (Hitachi, E-1010) and used for SEM (Hitachi, S-4800) observation.

### Immunofluorescence assay

Zebrafish primitive gonads or sections of testis were fixed with 4% paraformaldehyde (PFA) for immunofluorescence staining. After washing with ice-cold PBS three times, the samples were blocked with blocking solution (0.1% trition-100, 1% BSA and 1% DMSO in PBS) for at least 1 h. The antibodies (1:500) were then mixed in blocking solution and used to incubate the samples overnight at 4 °C. After staining with DAPI and washing with PBS, the samples were used for fluorescence microscopy imaging. The following antibodies were used for immunofluorescence staining. Anti-Vasa, anti-Sycp3 and anti-PCNA were against the antigens, and purified using antigen-affinity chromatography by our lab. These antibodies were conjugated with Alexa Fluor 488, 568 and 680 on demand using Alexa Fluor® Antibody Labeling Kits according to the manufacturer's manual (Thermo Fisher). The effectiveness and specificity of antibodies have been described in a previous study[68].

### CRISPR/Cas9-mediated genome KI

For CRISPR/Cas9-mediated genome KI, MMEJ-mediated gene KI was used in the genomic loci of *mpx* and *sox19b*, and the gRNA target sequences (*mpx*-gRNA, CCTTCAAAGATCACCCTCCATGG, and *sox19b*-gRNA, TGCCCGGAGGAGACATGCCCGGG) were located at exon 10 of *mpx* and exon 1 of *sox19b*, respectively. Microhomology (6 bp) within the gRNA target was utilized to facilitate microhomology-mediated integration. The recovery of genome-located gRNA targets after precise integration was prevented by introducing synonymous mutations downstream of gRNA on targeted vectors. The NHEJ-mediated gene KI approach[52] was used in the genomic loci of *nanog*, and the gRNA target sequence, TGGGAGTAAATGGCACTCCAGGG, was on the last intron. All the donor vectors contained the missing CDS and were linked to EGFP or mCherry via P2A.

### smFISH

smFISH was performed essentially as described[42]. Briefly, embryos fixed with 4% PFA were prehybridized for 30 min at 37 °C in a hybridization buffer (30% formamide, 5 X SSC, 9 mM citric acid, 0.1% Tween 20, 50 μg/ml heparin, 10% dextran sulfate, and 1 X Denhardt's solution) after being washed with PBS containing 0.1% Tween20 and permeabilized with 0.5%tritionX-100 and protease K. The embryos were then transferred to another hybridization solution containing a mixture of 10 pmol split-initiator probes, and incubated overnight at 37 °C. After hybridization, the sections were washed with wash buffer (30% formamide, 5 X SSC, 9 mM citric acid, 0.1% Tween-20 and 50 μg/ml heparin) at 37 °C, followed by three washes for 10 min in 5 X SSCT at room temperature (RT). Next, embryos were incubated in amplification buffer (10% dextran sulfate in 5 X SSCT) with 30 nM hairpin DNA pairs overnight in the dark at RT. Finally, the embryos were mounted with 70% glycerol-PBS gradient.

### qRT-PCR

The total RNA was extracted from the gonads of iPGCT and WT (3 mpf) zebrafish using the TRIZOL method (Invitrogen). cDNA was synthesized using PrimeScript™ RT Reagent Kit (Takara). The primers used are listed in Supplementary Table 1. PCR amplification was performed using SYBR Green PCR reagent (Bio-Rad). The relative expression level (ΔCt) of each gene in every sample was calculated by normalizing the minimal cycle threshold (Ct) to the expression of β-actin[69]. There were three technical replicates for each sample, and their average were used to construct heatmaps to display the relative expression levels of multiple genes at the same time[70].

### Statistical analysis

iPGCT transplantation was performed using CellTram 4r Oil (Eppendorf) instruments, the transplanting needle was made using Flaming/Brown type micropipette puller (P-100, Sutter) and the needle was polished using Micro Grinder (EG-400, NARISHIGE). Testis and early embryos were imaged using a fluorescence stereomicroscope (Axio Zoom.V16, Zeiss), while sections and primitive gonads were imaged using a confocal microscope (SP8, Leica). The migration of iPGCs was tracked using a fluorescence microscope (CTR6500, Leica). The schematic drawing (Figs. 1a, c, j, 2a, 3a, b, g, e, 4a, 5e and Supplementary Figs. 1i and 3a) were drawn using Microsoft Office PowerPoint. Immunostaining experiments were repeated at least three times and representative example are shown. Images were processed using Image J software, and data were analyzed using GraphPad Prism 8.0 software. All data are presented as mean values ± SEM. Unpaired two-tailed Student's *t*-test was used to calculate the *P* values.

### Reporting summary

Further information on research design is available in the Nature Portfolio Reporting Summary linked to this article.

## Data availability

All data generated in this study, which include original data and images, are provided in the Supplementary Information. Source data are provided with this paper.

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

## Acknowledgements

We sincerely thank Luyuan Pan and Linglu Li of the China zebrafish Resource Center of the National Aquatic Biological Resource Center for their assistance in in vitro fertilization. We thank Fang Zhou, Yuan Xiao and Guangxin Wang of the analytical and testing center of Institute of Hydrobiology, CAS for the support of confocal imaging and scanning electron microscopy. This work was supported by the National Natural Science Foundation of China (32025037, 31721005, and 31902364), the Ministry of Science and Technology of China (2018YFA0801000), the Chinese Academy of Sciences (XDA24010108), and the Ministry of Agriculture and Rural Affairs of China (NK2022010207) to Y.S. X.W. was supported by the China Postdoctoral Science Foundation (E11Z0305), the Hubei Province Postdoctoral Innovation Research Post (E1390303), and the State Key Laboratory of Freshwater Ecology and Biotechnology (2021FB09). Y.S. was supported by the Qianjiang Scholar Program.

## Author contributions

Y.S. conceived the idea and oversaw the project. X.W. designed and performed the experiment and analyzed the data. J.Z. participated in most of the experiments. H.W., W.D., S.J. and M.H. participated in the genome-editing experiments. Y.W. performed the single molecule in situ hybridization experiment. F.Z. constructed the germplasm expression vectors. T.L., Y.H. and D.Y. performed partial experiments on Gobiocypris rarus. X.W. and Y.S. wrote the manuscript. All authors discussed the results and commented on the manuscript.

## Competing interests

The authors declare no competing interests.
