## [Peer Review File · Nature Communications]

Induced formation of primordial germ cells from zebrafish blastomeres by germlasm factorsEditorial Note: Parts of this Peer Review File have been redacted as indicated to maintain the confidentiality of unpublished data.

REVIEWER COMMENTS

Reviewer #1 (Remarks to the Author):

In this study, Wang et al demonstrate the induction of primordial germ cells (PGCs) in zebrafish by introducing 9 germplasm (9GM) mRNA into any blastula cells. 9GM-induced PGCs (iPGCs) can produce functional sperm upon transplantation into PGC-depleted host embryos. The authors also demonstrate that the same cocktail works in blastula of *Gobio cypris rarus* (Gr), a cyprinid fish. Xenotransplantation between Gr-derived iPGCs and zebrafish host increases the efficiency of Gr-derived sperm production compared with the conventional method. Finally, they apply their iPGC technology to generate genome-edited zebrafish. Overall, these results are interesting, and the manuscript is well-written. The finding would be practically beneficial for the production of genetically modified zebrafish, and potentially for the other species that generate PGCs by preformation manner.

One major interest that has not been addressed in this study is which of 9GM are crucial for PGC induction as the authors discuss in lines 223-228. The experiment to identify a combination of key germplasm for PGC induction in zebrafish would strengthen the current study.

In addition, I have several minor comments below.

Minor comments

1. Line 17 in the abstract, what does “mosaic overexpression” mean? As far as I read the main text, the authors do not mention the expression pattern of 9GM.

2. In lines 78-81 and Figure 1e, the authors claim that Vasa positive cells (green) are iPGCs. However, this cannot distinguish transplanted iPGCs and residual PGCs in host embryos. The authors should show non-transplanted control to clarify and morpholino completely depletes endogenous PGCs. If the GFP expression is still observed at 10 dpf, double staining by Vasa and GFP would be informative to clarify whether all the GFP positive iPGCs express Vasa or not.

3. Throughout the manuscript, the authors compare transplantation efficiency between iPGCT and conventional PGCT. Do they transplant the same number of donor cells both in iPGCT and PGCT? I could not find their detail in the method section. In the discussion (lines 220-221), the authors mention that 1 donor is transplanted to 2-3 hosts in PGCT, but 1 donor to 30 hosts in iPGCT. If so, the comparison should not be fair because the number of the cell transplanted is different, which affect the estimation of the efficiency. The authors should clarify it in the revised manuscript.

4. I am a bit confused about efficiency in Figure 1f and Table1. I guess “PGCs migrate to genital ridge” in Y axis (Figure 1f) and column (Table1) means the percentage/number of “embryos with PGCs that migrated to genital ridge”. If so, the authors should correct it. Otherwise, the reader may confuse about how to estimate the efficiency.

5. In Figure 1f, while the efficiency of control PGCT is the same between 1 and 4 dpf, that of

iPGCT seems significantly decrease from 1 to 4 dpf. The percentage of efficiency in iPGCT at 4 dpf is almost corresponding to the percentage of fertile adults in Table 1, suggesting some GFP positive cells at 1 dpf or earlier stage are non-PGCs that fail to activate endogenous PGC genes. Thus, it should be clarified how much percentage of the GFP positive cells can activate endogenous PGC genes, and when they initiate their expression.

6. In Table 1, the authors should add a column showing the number of embryos transplanted in each test. Also, they should add information when they assess "PGCs migrate to genital ridge", which I guess 1 dpf. Adding how to calculate the percentage (eg. x/y , y/z) would be informative for the reader.

7. In lines 166-170, it is not clear why many xenografts lost during PGC development compared with allografts. The authors should add this point to the discussion.

8. Distribution of panels in each figure is difficult to follow (eg. In Figure 2, next to c is somewhat h, f, g, then d, and e are followed by). I recommend the authors distribute the panels more sequential manner.

9. Line 189, Figure 6e should be Figure 5e. Other than that, there are several mislabeling of figure numbers. Please carefully double-check them again.

Reviewer #2 (Remarks to the Author):

The manuscript by Wang et al presents a method by which somatic cells of fish embryos (demonstrated primarily in zebrafish) are induced to become primordial germ cells (PGCs) by expression of RNA molecules encoding for 9 germ-plasm proteins. The induced PGCs (iPGCs) are then transplanted into host zebrafish embryos and were shown to express vasa as a germ-cell marker. The iPGCs develop to become functional gametes as evidenced by the offspring of the host fish. Genetic editing was carried out on the iPGCs ; specifically, in a clonal population of iPGCs, 3 genes were knocked out generating a chimeric germline, and knock in of GFP and CDS sequences was carried out in the entire iPGC population which were then transplanted into a panel of hosts. Last, iPGCs, derived from one fish species, generated by using the same 9-factor cocktail, were then allowed to deveop in zebrafish hosts.

The method described in the paper does not provide new insights into germ-cell specification and later development, nor into the development of somatic tissues. While the procedure offers higher throughput when conducting certain procedures, it is conceptually not new. Specific comments.

1. The basis for choosing the 9 components is not clear. Are all of them needed? are the RNA concentrations optimized in any way? For a new method, this information would be required.

2. What is special about the iPGCs that does not support the development of female fish? The possible explanation provided by the authors "In iPGCT embryos, apoptosis or immune rejection of donor iPGCs can lead to a stronger immune response and eventually female-to-male reversal. " It is not clear how intraspecies transplantation (the animals are inbred) leads to rejection. Why only of oocytes.

3. What is the fate of the progeny of animals that were obtained from the iPGCs? This point is important if the method is used for the generation of adult lines carrying transgenes and mutations. Similarly, using the procedure for maintaining wild-type fish for species

preservation and agriculture would require the generation of healthy, fertile, and normal adults.

4. The methods section and the figure legends are very minimal and provide very little information regarding the precise experimental design and details. Similarly, the movies and tables are not accompanied by legends. What is the context of the green cells in Fig 1e, 10d embryo. There are also multiple mistakes. For example in Fig 2c itself the stage is 32dpf, while the legend states that the stage is 25dpf. Also, in Figure 2d, it is not clear from the panel what the difference between normal testes (Tg(cmv:GFP)) and the dnd MO. What was the protocol used for the qRT PCR? Another example is in Fig 2m which has 12 panels with a minimal description.. 'Characterization of the recombinant gonads by immunofluorescence'...what is 1 and what is 2. Magnification?
5. It would recommend following other germ-cell characteristics such as germ-cell granules. Are all the cells shown in Fig 1B show perinuclear granules?
6. The statement that "In contrast to ePGCs, which generally divided slowly, blastula cell-derived iPGCs still divided rapidly in host embryos, mimicking early embryonic cells (Movie 2)" is not observed in the movie and there is no quantitation of the division rate of the iPGCs vs the ePGCs.
7. Movie 4 could be shortened and the efficiency it should show is not obvious from the presentation.
8. The actual phenotypes induced by CRISPR should be presented.

Reviewer #3 (Remarks to the Author):

The manuscript by Wang et al. describes the novel strategy, using induced primordial germ cells (iPGCs) *in vivo*, to generate knocking-out (KO) or knocking-in (KI) zebrafish. The authors demonstrate that ectopic expression of 9 germline factors (9GM) in the early zebrafish embryo is sufficient to make somatic cells to convert into the germ cell lineage, thereby making iPGCs *in vivo*. In combination of authentic genome editing strategies, transplantation of iPGCs in wild-type host embryos allows for efficient generation of KO or KI zebrafish. Furthermore, using the same strategy, zebrafish donor iPGCs can be used for genetic manipulations of another teleost fish, *Gobiocypris rarus* (Gr).

Generation of iPGCs in zebrafish *in vivo* is novel, and is highly applicable for *in vivo* genome editing, in particular for genes that are required for early development. Thus, this work can lead to a paradigm shift for genome editing in zebrafish. I would support this work for publication in Nature Communications if my concerns were clarified.

(Major points)

1. Figure 3f require more experiments to increase the number of the success cases. Just showing one case each for pou5f3 or tyr with low efficiency (10%) is not convincing.
2. The detailed design and procedure of the KI strategy should be shown in supplemental Figure and materials & methods, including how long homologous sequences is used on either the 5'- or 3'- region and all the sequences of gRNAs (sox19b and nanog). It seems that 'NHEJ' is used for nanog knock-in as well as 'MMEJ' for mpx and sox19b. This should be described in materials & methods, as well.

(Minor points)

Line 68, Reference 11 is not a proper article to cite. Instead, here, the authors should cite the original paper for each of 9 germline factors separately.

Line 101. , 'asymmetric migration of iPSCs' is unclear.

Figure 2m needs proper labeling of '1'.

Line 189, 'As shown in Figure 6e,...' but I cannot find the figure.

Line 412, Figure legend1 (b), it should be ...buc or 9GM mRNA to induce PGCs, instead of buc and 9GM mRNA.

Line 414, Spell induced PGC transplantation (iPGCT), instead of PGCT.

Line 416. The legend for (e) requires how to visualize the PGC-specific protein Vasa (immunohistochemistry or GFP reporter?).

Line 417, p-values need to be shown or in the corresponding main text.

Line 427, '... meiosis and meiosis...' should be '...meiosis and mitosis...'.

Line 434, , the p-values need to be shown or in the corresponding main text.

Line 473, 'MMEJ' needs to be spelled in full.

Revision summary and Point-by-Point Response to Reviewers

1. Summary of major comments from the reviewers.

Based on the requests and comments from the reviewers, we have performed further analyses and additional experiments. Our major revision and new supporting data are summarized and listed in the following Table 1.

Table 1. Revision summary and new supporting data to address major comments from the reviewers.

	Questions	Reviewers	Clarification on the original submission data and New data supporting
1	One major interest that has not been addressed in this study is which of 9GM are crucial for PGC induction as the authors discuss in lines 223-228.	1#: Q1	Thank you! We totally agree with you on this point! In fact, we initially used 13GM, including 9GM related to PGC specialization (buc, piwil1, nanos3, ddx4, dazl, dnd, tdrd6, tdrd7, dazap2) and 4GM related to PGC migration (rgs14a, cxcr4a, cxcr4b, ca15b) (Response New Figure 1) for PGC induction experiment, and we screened out 9GM from 13GM. We have added the relevant data into the revised manuscript Figure 1b (13GM) and Supplementary Fig 1b (see also Response new Figure 2a, b). When we removed the 4GM related to PGC migration, the remaining 9GM were still able to induce PGC efficiently (Revised manuscript Fig. 1b-h, Supplementary Movie 1), indicating that the 4GM related to PGC migration do not belong to the key germplasms for PGC induction. These results have been added in the revised manuscript (P5-6, L73-83, L94-99). According to your suggestion, we also removed the factors in 9GM one by one, and finally identified a combination of germplasms (nGM) containing less factors, which was capable of efficient PGC induction (Response New Figure 3a-f, Figure for Reviewer 1 only). Since all the experiments in the current manuscript have been performed with 9GM, we only presented the data related to 9GM in this manuscript. The underlying mechanism of nGM inducing PGC is still under careful investigation, thus this piece of work will be certainly presented in a future manuscript.

2	Line 17 in the abstract, what does “mosaic overexpression” mean? As far as I read the main text, the authors do not mention the expression pattern of 9GM.	1#: Q2	The original “mosaic overexpression” means localized injection of 9GM mRNAs into any blastomere. However, given the word limit of the abstract, we have revised the text and hope that it is now clearer (P1, L15-18).
3	In lines 78-81 and Figure 1e, the authors claim that Vasa positive cells (green) are iPGCs. However, this cannot distinguish transplanted iPGCs and residual PGCs in host embryos. The authors should show non-transplanted control to clarify dnd morpholino completely depletes endogenous PGCs. If the GFP expression is still observed at 10 dpf, double staining by Vasa and GFP would be informative to clarify whether all the GFP positive iPGCs express Vasa or not.	1#: Q3	According to your advice, we added the relevant results that were missing from the original manuscript (P6-7, L114-119). First, we added relevant data of non-transplanted control to clarify that dnd morpholino completely depleted endogenous PGCs (Revised manuscript Supplementary Fig 1f, g, also see Response New Fig. 4a, b). Second, we identified the transplanted iPGC with GFP expression (injected with GFP-UTRnanos3 mRNA), and stained the iPGC with Vasa antibody, to distinguish the transplanted iPGCs from residual PGCs (without GFP expression) (Revised manuscript Fig. 1i, also see Response New Fig. 4c). It is true that all the GFP-positive iPGCs express Vasa.
4	Throughout the manuscript, the authors compare transplantation efficiency between iPGCT and conventional PGCT. Do they transplant the same number of donor cells both in iPGCT and PGCT? I could not find their detail in the method section. In the discussion (lines 220-221), the authors mention that 1 donor is transplanted to 2-3 hosts in PGCT, but 1 donor to 30 hosts in iPGCT. If so, the comparison should not be fair because the number	1#: Q4	In fact, in conventional PGCT and iPGCT, the number of donor cells used for transplantation is not the same, i.e., 50-100 cells were transplanted in conventional PGCT, and 10-20 cells were transplanted in iPGCT. But this doesn't mean that the comparison is unfair. The reason is presented as below. In conventional PGCT, “1 donor could be maximumly transplanted to 2-3 hosts in PGCT” means one donor's PGCs (about 8 PGCs) can be theoretically transplanted to up to 2-3 embryos through PGCT. In practice, it is even impossible to accurately aspirate the PGCs from donor embryo at 30% epiboly (about 8 PGCs, Revised manuscript Fig. 1e, f, also see Response New Fig. 5a, b), so the success rate of transplantation through PGCT is about 10%. This was described in a previous study (Ciruna al et. , PNAS, 2002, doi: 10.1073/pnas.222459999) and also confirmed by our

	of the cell transplanted is different, which affect the estimation of the efficiency. The authors should clarify it in the revised manuscript.		experiment (Revised manuscript Fig. 1g, h, also see Response New Fig. 5c, d). Therefore, in conventional PGCT, a relative high amount of donor cells (50-100 cells) was generally transplanted to a recipient embryo, in order to increase the chance of obtaining donor-derived PGCs. Unlike conventional PGCT, a less amount (10-20 cells) of donor cells was required for iPGCT after optimizing the procedure, because almost all blastoderm cells were induced into iPGC by 9GM, and the transplanted iPGCs even have a faster proliferation rate (Revised manuscript Fig. 1e-h, Supplementary Fig. 1c, Supplementary Movie 2, also see Response New Fig. 5a-e). We have added the details into the Method section and revised the Discussion part (Revised manuscript Fig. 1e-h, Supplementary Fig. 1c, also see Response New Fig. 5 and P6, L99-110; P14-15, L284-292; P16-17, L326-337).
5	I am a bit confused about efficiency in Figure 1f and Table1. I guess “PGCs migrate to genital ridge” in Y axis (Figure 1f) and column (Table1) means the percentage/number of “embryos with PGCs that migrated to genital ridge”. If so, the authors should correct it. Otherwise, the reader may confuse about how to estimate the efficiency.	1#: Q5	We have corrected it (see revised manuscript Fig. 1h and Table 1, also see Response New Fig. 5d, New Table 1).
6	In Figure 1f, while the efficiency of control PGCT is the same between 1 and 4 dpf, that of iPGCT seems significantly decrease from 1 to 4 dpf. The percentage of efficiency in iPGCT at 4 dpf is almost corresponding to the percentage of fertile adults in Table 1, suggesting some GFP	1#: Q6	Indeed, the percentage of efficiency in iPGCT at 4 dpf is almost corresponding to the percentage of fertile adults in original Table 1. The numbers of iPGCs in the genital ridge of 1 dpf embryos varied a lot, and a certain proportion of the embryos with low amount iPGCs tended to develop to PGC-less embryos at 4 dpf and finally into infertile adults (Revised manuscript Fig. 1e-h, Table 1, also see Response New Fig. 5a-d, and New Table 1), suggesting a sufficient of iPGCs at 1 dpf is necessary for the success of iPGCT. In order to make Table 1 more informative, we have added more data into

	positive cells at 1 dpf or earlier stage are non-PGCs that fail to activate endogenous PGC genes. Thus, it should be clarified how much percentage of the GFP positive cells can activate endogenous PGC genes, and when they initiate their expression.		this table, and change the column title accordingly (P6, L106-110). To clarify how much percentage of the GFP positive cells can activate endogenous PGC genes and when they initiate their expression, we conducted additional experiment by visualizing neonatal germplasm mRNA. The neonatal mRNA of tdrd7a and ddx4 was detected by single molecule in situ hybridization (smFISH) technology. To distinguish the endogenously transcribed mRNA (neonatal mRNA containing endogenous 3'UTR) from the exogenously loaded mRNA (without endogenous 3'UTR), we designed probes in the UTR regions of mRNA (en_probes) and CDS (ex_probes), respectively (Revised manuscript Supplementary Fig. 1h, i, also see Response New Fig. 6a, b). ex_probes signals of tdrd7a and ddx4 were detected from 30% epiboly to 5-somite stage, while en_probes signals were detected only at 5-somite stage and 1 dpf, and almost all iPGCs could activate endogenous PGC genes (Revised manuscript Supplementary Fig. 1h, i and Fig. 1j, also see Response new Fig. 5a-c). This indicates that iPGC has initiated the expression of germplasm genes at 5-somite stage. All these data suggest that iPGCs induced by 9GM could activate endogenous PGC genes from 5-somite stage, and we have added these results in the revised manuscript (P7, L119-130).
7	In Table 1, the authors should add a column showing the number of embryos transplanted in each test. Also, they should add information when they assess "PGCs migrate to genital ridge", which I guess 1 dpf. Adding how to calculate the percentage (eg. x/y, y/z) would be informative for the reader.	1#: Q7	We have added the details for calculating the percentages (see revised manuscript Table1, also see Response New Table 1).
8	In lines 166-170, it is not clear why many xenografts lost during PGC development compared with allografts. The authors should add this point to the	1#: Q8	Immune rejection of Gobiocypris rarus (Gr) iPGCs by zebrafish may be the main reason for the low success rate of Gr_iPGCT_Dr. On the other hand, germplasm factors derived from zebrafish were used for iPGC of Gobiocypris rarus, likely leading to low efficiency of xenografts. We added a discussion to the Revised

	discussion.		manuscript (P15, L302-305).
9	Distribution of panels in each figure is difficult to follow (eg. In Figure 2, next to c is somewhat h, f, g, then d, and e are followed by). I recommend the authors distribute the panels more sequential manner.	1#: Q9	Sorry for the confusion. We have repositioned the panels in order in each figure.
10	Figure 6e should be Figure 5e. Other than that, there are several mislabeling of figure numbers. Please carefully double-check them again.	1#: Q10	Sorry for these mistakes. We have corrected this mistake and all the other mistakes of mislabeling.
11	The method described in the paper does not provide new insights into germ-cell specification and later development, nor into the development of somatic tissues. While the procedure offers higher throughput when conducting certain procedures, it is conceptually not new.	2#: Q11	We agree with you on this point that the original manuscript does not provide enough new insights into the mechanism of germ cell specification. In the manuscript, however, we mainly focused on developing this novel method of PGC induction and iPGC transplantation for conducting high throughput experiment that requires PGC transplantation, which is generally with extremely low efficiency and low throughput. By using this special system, we also preliminarily explored the mechanism of PGC specification. The induction of fish PGCs has been realized through overexpression of Buc, which is able to recruit germ plasm components (Bontems F, et al., Current Biology, 2009, doi: 10.1016/j.cub.2009.01.038). However, this induction method only doubles the number of PGCs in early embryo (Bontems F, et al., Current Biology, 2009, doi: 10.1016/j.cub.2009.01.038; Zhang, et al., Journal of Genetics and Genomics, 2020, doi: 10.1016/j.jgg.2019.12.004). In this manuscript, by identifying 9 germplasm factors (9GM) from 13 GM, we could even induce almost the entire blastoderm cells to PGCs, which should be a new paradigm for PGC induction and PGC transplantation in teleost fish. The highly efficient iPGCT approach developed by our study is useful not only for editing embryonic lethal genes in a high throughput manner, but also for efficient breeding

		of aquaculture species by genome editing. In an effort to provide certain new insights into PGC specification, we conducted additional experiments, including analyzing neonatal mRNA of PGC genes, and added the data related to screening out 9GM from 13GM. The relevant experiment and results are briefly present as below. First, we conducted additional experiment by visualizing neonatal germplasm mRNA the transplanted iPGCs, in order to clarify whether the iPGCs can activate endogenous PGC genes and when they initiate their expression. The neonatal mRNA of tdrd7a and ddx4 was detected by single molecule in situ hybridization (smFISH) technology. To distinguish the endogenously transcribed mRNA (neonatal mRNA containing endogenous 3'UTR) from the exogenously loaded mRNA (without endogenous 3'UTR), we designed probes in the UTR regions of mRNA (en_probes) and CDS (ex_probes), respectively (Revised manuscript Supplementary Fig. 1h, i, also see Response New Fig. 7a, b). ex_probes signals of tdrd7a and ddx4 were detected from 30% epiboly to 5-somite stage, while en_probes signals were detected only at 5-somite stage and 1 dpf, and almost all iPGCs could activate endogenous PGC genes (Revised manuscript Supplementary Fig. 1h, i and Fig. 1j, also see Response New Fig. 7a-c). This indicates that iPGC has initiated the expression of germplasm genes at 5-somite stage. All these data suggest that iPGCs induced by 9GM could activate endogenous PGC genes from 5-somite stage, and we have added these results in the revised manuscript (P7, L119-130). Second, in previous studies, although Buc could recruit existing nearby germplasm factors to induce ectopic PGCs, the core factors that are sufficient for inducing PGCs are not well defined. In this manuscript, we screened out 9GM from 13GM, and the localized injection experiment strongly suggest that the 9GM are sufficient to induce any blastoderm cell into PGC (Revised manuscript Fig. 1b, g, h, Supplementary Fig. 1b and revised manuscript Fig. 3a, b, also see Response New Fig. 8, Fig. 9a-f). We initially used 13GM, including 9GM related to PGC specialization (buc, piwil1, nanos3, ddx4, dazl, dnd, tdrd6, tdrd7,
--	--	---

			dazap2) and 4GM related to PGC migration (rgs14a, cxcr4a, cxcr4b, ca15b) (Response New Fig. 8) for PGC induction experiment, and we screened out 9GM from 13GM. We have added the relevant data into the revised manuscript Fig. 1b (13GM) and Supplementary Fig. 1b (see also Response new Fig. 9a-d). When the 4GM related to PGC migration were removed, the remaining 9GM were still able to induce PGC efficiently, indicating that the 4GM related to PGC migration do not belong to the key germplasms for PGC induction. These results have been added in the revised manuscript (P5, L73-83, L94-99; P9, L169-178). Third, in our ongoing project, we continued to optimize the PGC induction system. 9GM was removed from the system one by one, and we finally identified a combination of germplasms (nGM) containing less factors, which was capable of efficient PGC induction (Response New Fig. 10a-f, Figure for Reviewer 2 only). Since all the experiments in the current manuscript have been performed with 9GM, we only presented the data related to 9GM in this manuscript. The underlying mechanism of nGM inducing PGC is still under careful investigation, thus this piece of work will be presented in a future manuscript.
12	The basis for choosing the 9 components is not clear. Are all of them needed?	2#: Q12-1	Thank you for pointing out this issue. In the revised manuscript, we added the data related to screening out 9GM from an original combination of 13GM. We initially used 13GM, including 9GM related to PGC specialization (buc, piwil1, nanos3, ddx4, dazl, dnd, tdrd6, tdrd7, dazap2) and 4GM related to PGC migration (rgs14a, cxcr4a, cxcr4b, ca15b) (Response New Fig. 8) for PGC induction experiment, and we screened out 9GM from 13GM. We have added the relevant data into the revised manuscript Fig. 1b (13GM) and Supplementary Fig. 1b (also see Response New Fig. 9a, b). When we removing the 4GM related to PGC migration, the remaining 9GM were still able to induce PGC efficiently (Revised manuscript Fig. 1g, h, also see Response New Figure 9c, d), indicating that the 4GM related to PGC migration do not belong to the key germplasms for PGC induction. These results have been added in the revised manuscript (P5, L73-83, L94-99).

			In our ongoing project, we continued to optimize the PGC induction system. 9GM was removed one by one, and finally identified a combination of germplasm (nGM) containing less factors, which was capable of efficient PGC induction (Response New Figure 10a-f, Figure for Reviewer 2 only). Since all the experiments in the current manuscript have been performed with 9GM, we only presented the data related to 9GM in this manuscript. The underlying mechanism of nGM inducing PGC is still under careful investigation, thus this piece of work will be presented in a future manuscript.
13	Are the RNA concentrations optimized in any way?	2#: Q12-2	Thank you for pointing out this issue. According to your advice, we added the relevant results that were missing from the original manuscript (P5, L83-88). Initially, we have optimized the RNA concentration. At low dose (25 pg per factor), GFP-UTRnanos3 was weakly positive. At high doses (100 pg per factor), the proliferation of embryonic cells was strongly inhibited. In contrast, iPGC can be effectively induced only at moderate dose (50 pg per factor) (Revised manuscript Supplementary Fig. 1a, also see Response New Fig. 11). Therefore, PGC induction was performed with moderate dose of germplasm factors in subsequent experiments.
14	What is special about the iPGCs that does not support the development of female fish? The possible explanation provided by the authors "In iPGCT embryos, apoptosis or immune rejection of donor iPGCs can lead to a stronger immune response and eventually female-to-male reversal." It is not clear how intraspecies transplantation (the animals are inbred) leads to rejection. Why only of oocytes.	2#: Q13	Thank you for pointing out this issue. To explore the reasons why the iPGCs that does not support the development of female fish, we continuously tracked the proliferation and differentiation of iPGCs (Revised manuscript Supplementary Fig. 2a-d, also see Response New Fig. 12a-d). From 12 dpf to 32 dpf, the number of germ cells in iPGCT fish was significantly lower than that in control zebrafish, and the developmental progress of germ cells in iPGCT fish was also obviously retarded when compared with the control (Revised manuscript Supplementary Fig. 2c, also see Response New Fig. 12c). The lower number of germ cells in juvenile fish might lead to final masculinization of the iPGCT zebrafish, since a sufficient number of germ cells is required for female development in zebrafish. We also observed that the iPGCs in a few iPGCT embryos could still develop into stage I oocytes at 50

			dpf (Revised manuscript Supplementary Fig. 2d, also see Response New Fig. 12d), but we do not know whether these transplants could finally develop into females. We have added the relevant data and revised the Discussion accordingly (Revised manuscript Supplementary Fig. 2a-d, P15, L295-309).
15	What is the fate of the progeny of animals that were obtained from the iPGCs? This point is important if the method is used for the generation of adult lines carrying transgenes and mutations. Similarly, using the procedure for maintaining wild-type fish for species preservation and agriculture would require the generation of healthy, fertile, and normal adults.	2#: Q14	Thank you for your suggestion! The progeny of animals obtained from the iPGCs were able to grow and reproduce normally, and we have added the data into the revised manuscript (Revised manuscript Fig. 2g, Fig. 4i, also see Response New Fig. 13a, b).
16	The methods section and the figure legends are very minimal and provide very little information regarding the precise experimental design and details. Similarly, the movies and tables are not accompanied by legends. What is the context of the green cells in Fig 1e, 10d embryo. There are also multiple mistakes. For example, in Fig 2c itself the stage is 32 dpf, while the legend states that the stage is 25dpf. Also, in Figure 2d, it is not clear from the panel what the difference between normal testes (Tg(cmv:GFP)) and the dnd MO. What was the protocol	2#: Q15-1	Sorry for these mistakes. We have corrected these mistakes and added detailed figure legends and movies legends.

	used for the qRT PCR? Another example is in Fig 2m which has 12 panels with a minimal description. 'Characterization of the recombinant gonads by immunofluorescence'...what is 1 and what is 2. Magnification?		
17	What is the context of the green cells in Fig 1e, 10d embryo.	2#: Q15-2	The green cells were labeled with GFP-UTR nanos3 and were derived from the donor iPGCs. We have added detailed figure legends in revised manuscript (Revised manuscript Fig. 1i and P26, L646-648).
18	in Figure 2d, it is not clear from the panel what the difference between normal testes (Tg(cmv:GFP)) and the dnd MO.	2#: Q15-3	Thank you for pointing out this issue. We have added anatomical figures of the gonads in order to see the differences between the different groups more clearly (Revised manuscript Supplementary Fig. 2e , also see Response New Fig. 14).
19	What was the protocol used for the qRT PCR?	2#: Q15-4	Sorry for missing this protocol. We have added the details in the figure legends and methods in the revised manuscript (P19, L387-395). Briefly, relative expression level (Δ Ct) of each gene in every sample was obtained by qRT-PCR according to previous studies (Livak and Schmittgen, Methods , 2001, doi: 10.1006/meth.2001.1262), and presented as a heatmap according to a previous report (Mao, et al., Stem cell reports , 2016, doi: 10.1016/j.stemcr.2016.09.002; Rahman, et al., Small , 2020, doi: 10.1002/smll.202000272).
20	It would recommend following other germ-cell characteristics such as germ-cell granules. Are all the cells shown in Fig 1B show perinuclear granules?	2#: Q16	Thank you for pointing out this issue. According to your suggestion, we followed other germ-cell characteristics, such as germ-cell granules, and transcription of neonatal germplasm mRNA to characterize the iPGC, and added the new data as follows. First, at shield stage and 8 dpf, we found that Vasa protein was granulated in the perinuclear region of iPGCs (Revised manuscript Supplementary Fig. 1e , Fig. 1i , also see Response New Fig. 15a, b). In addition, tdrd7a and ddx4 mRNA showed perinuclear granule-specific distribution in almost all iPGCs at 5-somites and 1 dpf stages (Revised manuscript Fig. 1j ,

			Supplementary Fig. 1i, also see Response New Fig. 15c, d). Second, to identify neonatal mRNA containing endogenous 3'UTR from the exogenously loaded mRNA without endogenous 3'UTR, we designed probes in the UTR regions of mRNA (en_probes) and CDS (ex_probes), respectively, and detected them by single molecule in situ hybridization (smFISH) technology. ex_probes signals of tdrd7a and ddx4 were detected from 30% epiboly to 5-somite stage, while en_probes signals were detected only at 5-somite stage and 1 dpf, and almost all iPGCs could activate endogenous PGC genes (Revised manuscript Supplementary Fig. 1h, i and Fig. 1j, also see Response New Fig. 15c-e). This indicated that iPGC has initiated the expression of germplasm genes at 5-somite stage and possessed germ-cell characteristics. We have added these results in the revised manuscript (P6-7, L114-130).
21	The statement that "In contrast to ePGCs, which generally divided slowly, blastula cell-derived iPGCs still divided rapidly in host embryos, mimicking early embryonic cells (Movie 2)" is not observed in the movie and there is no quantitation of the division rate of the iPGCs vs the ePGCs.	2#: Q17	Thank you for pointing out this issue. According to your suggestion, we have done additional experiment, and added quantitation of the division rate of the iPGCs vs the ePGCs in revised manuscript Fig.1e, f, Supplementary Fig 1c, Supplementary Movie 2, also see Response New Fig. 16a-c. At 30% epiboly stage, around 10 iPGCs were transplanted to the recipient embryo. At 10-somites stage, the number of iPGC reaches around 80, which is much higher than the number of ePGC (an average of 20) (Revised manuscript Fig.1e, f, Supplementary Fig. 1c, Supplementary Movie 2, also see Response New Fig. 16a-c). The fast division of iPGCs was presented as a time-lapse panel in revised manuscript Supplementary Fig. 1c (also see Response New Fig. 16c). This indicates that iPGC may have the characteristics of rapid proliferation similar to somatic cells in the early stage, and gradually change to PGC characteristics in the somite stage. This is consistent with the expression of endogenous germplasm factors at 5-somites stage (Revised manuscript Fig.1j, Supplementary Fig 1h, i, also see Response New Fig. 15c-e). We have added results to the revised manuscript (P6, L99-110).

22	Movie 4 could be shortened and the efficiency it should show is not obvious from the presentation.	2#: Q18	Thank you! We've shortened the movie (Revised manuscript Supplementary Movie 3). This video mainly shows the iPGCT operation process, in which a large number of iPGCs could be aspirated at one time, and 10-20 cells were transplanted to each host embryo. Therefore, one donor embryo could provide iPGC for at least 30 host embryos, highlighting the characteristics of high throughput by iPGCT.
23	The actual phenotypes induced by CRISPR should be presented.	2#: Q19	Thank you for pointing out this issue. We have added the relevant data into the Revised manuscript Supplementary Fig. 3b-d, also see Response New Fig. 17a-c.
24	Figure 3f require more experiments to increase the number of the success cases. Just showing one case each for pou5f3 or tyr with low efficiency (10%) is not convincing.	3#: Q20	Thank you for pointing out this issue. We have conducted additional experiments to show the high efficiency of iPGC-targeted genome-editing by completely removing endogenous PGCs. The new data was added in Revised manuscript Fig. 3g, h (also see Response New Fig. 18a, b), and revised main text (P10, L196-202). Briefly, in order to remove endogenous PGCs, we now injected dnd MO2 (ATGTCTCCGACCATCTGTGATGATG, Gross-Thebing et al., Developmental Cell, 2017, doi: 10.1016/j.devcel.2017.11.019), which could inhibit the translation of endogenous dnd mRNA, but not exogenous dnd mRNA. In this experiment, the efficiency of gametic mutation is greatly improved.
25	The detailed design and procedure of the KI strategy should be shown in supplemental Figure and materials & methods, including how long homologous sequences is used on either the 5'- or 3'-region and all the sequences of gRNAs (sox19b and nanog). It seems that 'NHEJ' is used for nanog knock-in as well as 'MMEJ' for mpx and sox19b. This should be	3#: Q21	Thank you for pointing out this issue. We have added detailed explanations in the revised manuscript methods (P18, L363-374) and Supplementary Table 3.

	described in materials & methods, as well.		
26	Line 68, Reference 11 is not a proper article to cite. Instead, here, the authors should cite the original paper for each of 9 germplasm factors separately.	3#: Q22	Thank you for pointing out this issue. We have cited the original paper for each of 9 germplasm factors separately (P5, L75-76).
27	Line 101. ,'asymmetric migration of iPGCs' is unclear.	3#: Q23	Thank you for pointing out this issue. "asymmetric migration of iPGCs" mean that iPGC migrated mainly to the genital ridge on one side of the recipient embryo. We have added a clearer explanation.
28	Figure 2m needs proper labeling of '1'.	3#: Q24	We have corrected it (see revised manuscript Fig. 2f).
29	Line 189, 'As shown in Figure 6e,...' but I cannot find the figure.	3#: Q25	Sorry for this mistake. It should be Figure 5e. We have corrected it.
30	Line 412, Figure legend1 (b), it should be ...buc or 9GM mRNA to induce PGCs, instead of buc and 9GM mRNA.	3#: Q26	Sorry for this mistake. We have corrected it (P26, L642-643).
31	Line 414, Spell induced PGC transplantation (iPGCT), instead of PGCT.	3#: Q27	Sorry for this mistake. We have corrected it.
32	Line 416. The legend for (e) requires how to visualize the PGC-specific protein Vasa (immunohistochemistry or GFP reporter?).	3#: Q28	Thank you for pointing out this issue. According to your suggestion, we stained the iPGC with Vasa antibody and showed GFP-UTR nanos3 at the same time. It was found that GFP positive cells showed Vasa antibody positive (Revised manuscript Fig. 1i, also see Response New Fig. 19a), suggesting that iPGC expressed PGC-specific marker. In addition, GFP-UTR nanos3 -positive ePGC was not detected after injecting dnd MO at 1 dpf embryos. In addition, Vasa antibody was not detected in dnd MO embryos at 8 dpf (Revised manuscript Supplementary Fig. 1f, g, also see Response New Fig. 19b, c). These results indicate that endogenous PGC is completely removed. We have added results to the revised manuscript (P6-7, L114-119).
33	Line 417, p-values need to be shown or in the	3#: Q29	We have added p-values to the revised manuscript.

	corresponding main text.		
34	Line 427, '... meiosis and meiosis...' should be '...meiosis and mitosis...'	3#: Q30	Thank you! We have corrected it (P28, L665-666).
35	Line 434, the p-values need to be shown or in the corresponding main text.	3#: Q31	Thank you! We have added p-values to the manuscript (P26-27, L648-652).
36	Line 473, 'MMEJ' needs to be spelled in full.	3#: Q32	We have added the full spelling of MMEJ (microhomology-mediated end-joining) (P12-13, L245-247).

2. Point-by-point responses to reviewers

Reviewer #1:

In this study, Wang et al demonstrate the induction of primordial germ cells (PGCs) in zebrafish by introducing 9 germplasm (9GM) mRNA into any blastula cells. 9GM-induced PGCs (iPGCs) can produce functional sperm upon transplantation into PGC-depleted host embryos. The authors also demonstrate that the same cocktail works in blastula of *Gobiocypris rarus* (Gr), a cyprinid fish. Xenotransplantation between Gr-derived iPGCs and zebrafish host increases the efficiency of Gr-derived sperm production compared with the conventional method. Finally, they apply their iPGC technology to generate genome-edited zebrafish. Overall, these results are interesting, and the manuscript is well-written. The finding would be practically beneficial for the production of genetically modified zebrafish, and potentially for the other species that generate PGCs by preformation manner.

Thank you very much for your appreciation and critical advices.

One major interest that has not been addressed in this study is which of 9GM are crucial for PGC induction as the authors discuss in lines 223-228. The experiment to identify a combination of key germplasm for PGC induction in zebrafish would strengthen the current study.

Response 1: Thank you! We totally agree with you on this point! In fact, we initially used 13GM, including 9GM related to PGC specialization (*buc*, *piwil1*, *nanos3*, *ddx4*, *dazl*, *dnd*, *tdrd6*, *tdrd7*, *dazap2*) and 4GM related to PGC migration (*rgs14a*, *cxcr4a*, *cxcr4b*, *ca15b*) (Response New Figure 1) for PGC induction experiment, and we screened out 9GM from 13GM. We have added the relevant data into the revised manuscript Figure 1b (13GM) and Supplementary Fig 1b (see also Response new Figure 2a, b). When we removed the 4GM related to PGC migration, the remaining 9GM were still able to induce PGC efficiently (Revised manuscript Fig. 1b-h, Supplementary Movie 1), indicating that the 4GM related to PGC migration do not belong to the key germplasms for PGC induction. These results have been added in the revised manuscript (P5-6, L73-83, L94-99).

According to your suggestion, we also removed the factors in 9GM one by one, and finally identified a combination of germplasms (nGM) containing less factors, which was capable of efficient PGC induction (Response New Figure 3a-f, Figure for Reviewer 1 only). Since all the experiments in the current manuscript have been performed with 9GM, we only presented the data related to 9GM in this manuscript. The underlying mechanism of nGM inducing PGC is still under careful investigation, thus this piece of work will be certainly presented in a future manuscript.

Response New Figure 1: An initial 13GM used for PGC induction in this study
(Adapted from Agueo et al, 2017; Marlow et al, 2015; Paksa & Raz, 2015)

The genes marked red are 9GM, and the genes marked blue are the other four genes in 13GM.

Response New Figure 2: Identification of 9GM from 13GM for PGC induction in zebrafish

(a) At 1-cell stage, zebrafish embryos were injected with 13GM or 9GM mRNA (50 pg per fraction) to induce iPGCs. GFP-UTR*nanos3* was used to label putative PGCs. The embryos were photographed at 90% epibody. (b) iPGCs migrated to the genital ridge of the recipient embryos.

[Redacted]

Response New Figure 3: Identification of nGM (n<9) from 9GM for PGC induction in zebrafish (Figure for Reviewer 1 only).

(a) PGC was induced by injection of 9 factors or (9-1) factors at 1-cell stage. GFP-UTR*nanos3* was used to label putative PGCs. (b) After the induction factors were removed one by one, the k germline factors (kGM) with the greatest influence on iPGC

induction were combined to perform iPGC induction. Finally, the combination of key germplasm for PGC induction was obtained (nGM). (c) The schematic diagram for iPGCT. (d) nGM could induce PGC efficiently, and nGM-induced iPGCs could still efficiently migrate to the genital ridge of host embryos. (e) A schematic of nGM mRNA injected individually or together in a single cell at the 128-cell stage. (f) At 128-cell stage, nGM mRNA were injected into a single cell alone or together. The migration of iPGC was observed at 5-somite stage and at 1 dpf.

Minor comments

1. Line 17 in the abstract, what does “mosaic overexpression” mean? As far as I read the main text, the authors do not mention the expression pattern of 9GM.

Response 2: The original “mosaic overexpression” means localized injection of 9GM mRNAs into any blastomere. However, given the word limit of the abstract, we have revised the text and hope that it is now clearer (P1, L15-18).

2. In lines 78-81 and Figure 1e, the authors claim that Vasa positive cells (green) are iPGCs. However, this cannot distinguish transplanted iPGCs and residual PGCs in host embryos. The authors should show non-transplanted control to clarify *dnd* morpholino completely depletes endogenous PGCs. If the GFP expression is still observed at 10 dpf, double staining by Vasa and GFP would be informative to clarify whether all the GFP positive iPGCs express Vasa or not.

Response 3: Thank you for pointing out this issue. According to your advice, we added the relevant results that were missing from the original manuscript (P6-7, L114-119). First, we added relevant data of non-transplanted control to clarify that *dnd* morpholino completely depleted endogenous PGCs (Revised manuscript Supplementary Fig 1f, g, also see Response New Fig. 4a, b). Second, we identified the transplanted iPGC with GFP expression (injected with *GFP-UTRnanos3* mRNA), and stained the iPGC with Vasa antibody, to distinguish the transplanted iPGCs from residual PGCs (without GFP expression) (Revised manuscript Fig. 1i, also see Response New Fig. 4c). It is true that all the GFP-positive iPGCs express Vasa.

Response New Figure 4: (a) iPGC was labeled with GFP-UTRnanos3. At 1 dpf GFP-UTRnanos3 was not detectable in embryos injected with *dnd* MO. (b). At 8dpf, Vasa antibodies were not detectable in embryos injected with *dnd* MO. (c) Immunofluorescence

staining was used to detect the PGC-specific protein Vasa in GFP-expressed iPGCs at 8 dpf host embryo.

3. Throughout the manuscript, the authors compare transplantation efficiency between iPGCT and conventional PGCT. Do they transplant the same number of donor cells both in iPGCT and PGCT? I could not find their detail in the method section. In the discussion (lines 220-221), the authors mention that 1 donor is transplanted to 2-3 hosts in PGCT, but 1 donor to 30 hosts in iPGCT. If so, the comparison should not be fair because the number of the cell transplanted is different, which affect the estimation of the efficiency. The authors should clarify it in the revised manuscript.

Response 4: Thank you for pointing out this issue. In fact, in conventional PGCT and iPGCT, the number of donor cells used for transplantation is not the same, i.e., 50-100 cells were transplanted in conventional PGCT, and 10-20 cells were transplanted in iPGCT. But this doesn't mean that the comparison is unfair. The reason is presented as below.

In conventional PGCT, "1 donor could be maximumly transplanted to 2-3 hosts in PGCT" means one donor's PGCs (about 8 PGCs) can be theoretically transplanted to up to 2-3 embryos through PGCT. In practice, it is even impossible to accurately aspirate the PGCs from donor embryo at 30% epiboly (about 8 PGCs, Revised manuscript Fig. 1e, f, also see Response New Fig. 5a, b), so the success rate of transplantation through PGCT is about 10%. This was described in a previous study (Ciruna *al et.*, PNAS, 2002, doi: 10.1073/pnas.222459999) and also confirmed by our experiment (Revised manuscript Fig. 1g, h, also see Response New Fig. 5c, d). Therefore, in conventional PGCT, a relative high amount of donor cells (50-100 cells) was generally transplanted to a recipient embryo, in order to increase the chance of obtaining donor-derived PGCs.

Unlike conventional PGCT, a less amount (10-20 cells) of donor cells was required for iPGCT after optimizing the procedure, because almost all blastoderm cells were induced into iPGC by 9GM, and the transplanted iPGCs even have a faster proliferation rate (Revised manuscript Fig. 1e-h, Supplementary Fig. 1c, Supplementary Movie 2, also see Response New Fig. 5a-e).

We have added the details into the Method section and revised the Discussion part (Revised manuscript Fig. 1e-h, Supplementary Fig. 1c, also see Response New Fig. 5 and P6, L99-110; P14-15, L284-292; P16-17, L326-337).

Response New Figure 5: A comparison of between PGCT and iPGC

(a) GFP-UTR*nanos3* was used to label PGC and ePGC. Since GFP-UTR*nanos3* could not label endogenous PGC at 30% epiboly stage, immunofluorescence against Piwil1 antibody was used to detect endogenous PGCs at 30% epiboly. (b) The number of PGCs in wild type (WT) and iPGC at different stages. * $P < 0.05$, ** $P < 0.01$, *** $P < 0.001$. (c) iPGCs migrated to the genital ridge of the recipient embryo. (d) iPGC could significantly improve the success rate of PGC transplantation. Error bars, s.d.; significance values * $P < 0.05$, ** $P < 0.01$, *** $P < 0.001$. (e) About 10 iPGCs were transplanted into *Tg(piwil1:egfp-UTRnanos3)* host, the endogenous PGCS (ePGC) were labeled with GFP (White arrow marked), and the iPGCs were labeled by mCherry-UTR*nanos3*. iPGC and ePGC proliferation were tracked after transplantation.

4. I am a bit confused about efficiency in Figure 1f and Table1. I guess “PGCs migrate to genital ridge” in Y axis (Figure 1f) and column (Table1) means the percentage/number of “embryos with PGCs that migrated to genital ridge”. If so, the authors should correct it. Otherwise, the reader may confuse about how to estimate the efficiency.

Response 5: Sorry for the confusion. We have corrected it (see revised manuscript Fig. 1h and Table 1, also see Response New Fig. 5d, New Table 1).

Response New Table 1: Efficiency of iPGC and PGCT

Transplantation strategies	PGC-positive embryos/survived embryos at 1 dpf (%)	PGC-positive embryos/survived embryos at 4 dpf (%)	Positive fertile adults/adults originated from PGC positive 4 dpf embryos (%)
--	--	---

	PGCT	13/128 (10.16%)	6/65 (9.23%)	4/4 (100%)
Dr		43/46 (93.48%)	23/37 (62.16%)	15/15 (100%)
	iPGCT	42/43 (97.67%)	23/36 (63.89%)	19/19 (100%)
		31/31 (100%)	17/30 (56.67%)	14/14 (100%)
	PGCT	2/91 (2.20%)	1/83 (1.20%)	1/1 (100%)
Gr		132/135 (97.78%)	38/65 (58.46%)	3/16 (18.75%)
	iPGCT	120/120 (100%)	49/77 (63.64%)	4/22 (18.18%)

5. In Figure 1f, while the efficiency of control PGCT is the same between 1 and 4 dpf, that of iPGCT seems significantly decrease from 1 to 4 dpf. The percentage of efficiency in iPGCT at 4 dpf is almost corresponding to the percentage of fertile adults in Table 1, suggesting some GFP positive cells at 1 dpf or earlier stage are non-PGCs that fail to activate endogenous PGC genes. Thus, it should be clarified how much percentage of the GFP positive cells can activate endogenous PGC genes, and when they initiate their expression.

Response 6: Thank you for your instruction. Indeed, the percentage of efficiency in iPGCT at 4 dpf is almost corresponding to the percentage of fertile adults in original Table 1. The numbers of iPGCs in the genital ridge of 1 dpf embryos varied a lot, and a certain proportion of the embryos with low amount iPGCs tended to develop to PGC-less embryos at 4 dpf and finally into infertile adults (Revised manuscript Fig. 1e-h, Table 1, also see Response New Fig. 5a-d, and New Table 1), suggesting a sufficient of iPGCs at 1 dpf is necessary for the success of iPGCT. In order to make Table 1 more informative, we have added more data into this table, and change the column title accordingly (P6, L106-110).

To clarify how much percentage of the GFP positive cells can activate endogenous PGC genes and when they initiate their expression, we conducted additional experiment by visualizing neonatal germplasm mRNA. The neonatal mRNA of *tdrd7a* and *ddx4* was detected by single molecule in situ hybridization (smFISH) technology. To distinguish the endogenously transcribed mRNA (neonatal mRNA containing endogenous 3'UTR) from the exogenously loaded mRNA (without endogenous 3'UTR), we designed probes in the UTR regions of mRNA (en_probes) and CDS (ex_probes), respectively (Revised manuscript Supplementary Fig. 1h, i, also see Response New Fig. 6a, b). ex_probes signals of *tdrd7a* and *ddx4* were detected from 30% epiboly to 5-somite stage, while en_probes signals were detected only at 5-somite stage and 1 dpf, and almost all iPGCs could activate endogenous PGC genes (Revised manuscript Supplementary Fig. 1h, i and Fig. 1j, also see Response new Fig. 5a-c). This indicates that iPGC has initiated the expression of germplasm genes at 5-somite stage. All these data suggest that iPGCs

induced by 9GM could activate endogenous PGC genes from 5-somite stage, and we have added these results in the revised manuscript (P7, L119-130).

Response New Figure 6: GFP positive iPGCs activate endogenous PGC genes

(a) Probes were designed at different positions of mRNA, and single molecule in situ hybridization (smFISH) was used to detect neonatal germplasm mRNA. Probes for detecting overexpressed mRNA (ex_probes) and probes for neonatal mRNA (en_probes) were designed in the CDS and UTR regions of the mRNA, respectively. ex_probe signals of *tdrd7a* and *ddx4* were detected at 30% epiboly, 75% epiboly and 5-somite stages, while en_probe signals were detected only at 5-somite stage. (c) Neonatal mRNAs of *ddx4* and *tdrd7a* mRNAs could be clearly detected in the iPGCs at 1 dpf.

6. In Table 1, the authors should add a column showing the number of embryos transplanted in each test. Also, they should add information when they assess “PGCs migrate to genital ridge”, which I guess 1 dpf. Adding how to calculate the percentage (eg. x/y, y/z) would be informative for the reader.

Response 7: Sorry for the confusion. We have added the details for calculating the percentages (see revised manuscript Table1, also see Response New Table 1).

7. In lines 166-170, it is not clear why many xenografts lost during PGC development compared with allografts. The authors should add this point to the discussion.

Response 8: Thank you for pointing out this issue. Immune rejection of *Gobiocypris rarus* (Gr) iPGCs by zebrafish may be the main reason for the low success rate of Gr_iPGCT_Dr. On the other hand, germplasm factors derived from zebrafish were used for iPGC of *Gobiocypris rarus*, likely leading to low efficiency of xenografts. We added a discussion to the Revised manuscript (P15, L302-305).

8. Distribution of panels in each figure is difficult to follow (eg. In Figure 2, next to c is somewhat h, f, g, then d, and e are followed by). I recommend the authors distribute the panels more sequential manner.

Response 9: Sorry for the confusion. We have repositioned the panels in order in each figure.

9. Line 189, Figure 6e should be Figure 5e. Other than that, there are several mislabeling of figure numbers. Please carefully double-check them again.

Response 10: Sorry for these mistakes. We have corrected this mistake and all the other mistakes of mislabeling.

Reviewer #2:

The manuscript by Wang et al presents a method by which somatic cells of fish embryos (demonstrated primarily in zebrafish) are induced to become primordial germ cells (PGCs) by expression of RNA molecules encoding for 9 germ-plasm proteins. The induced PGCs (iPGCs) are then transplanted into host zebrafish embryos and were shown to express vasa as a germ-cell marker. The iPGCs develop to become functional gametes as evidenced by the offspring of the host fish. Genetic editing was carried out on the iPGCs ; specifically, in a clonal population of iPGCs, 3 genes were knocked out generating a chimeric germline, and knock in of GFP and CDS sequences was carried out in the entire iPGC population which were then transplanted into a panel of hosts. Last, iPGCs, derived from one fish species, generated by using the same 9-factor cocktail, were then allowed to develop in zebrafish hosts.

The method described in the paper does not provide new insights into germ-cell specification and later development, nor into the development of somatic tissues. While the procedure offers higher throughput when conducting certain procedures, it is conceptually not new.

Response 11: We agree with you on this point that the original manuscript does not provide enough new insights into the mechanism of germ cell specification. In the manuscript, however, we mainly focused on developing this novel method of PGC induction and iPGC transplantation for conducting high throughput experiment that requires PGC transplantation, which is generally with extremely low efficiency and low throughput. By using this special system, we also preliminarily explored the mechanism of PGC specification.

The induction of fish PGCs has been realized through overexpression of Buc, which is able to recruit germ plasm components (Bontems F, *et al.*, Current Biology, 2009, doi: 10.1016/j.cub.2009.01.038). However, this induction method only doubles the number of PGCs in early embryo (Bontems F, *et al.*, Current Biology, 2009, doi: 10.1016/j.cub.2009.01.038; Zhang, et al., Journal of Genetics and Genomics, 2020, doi: 10.1016/j.jgg.2019.12.004). In this manuscript, by identifying 9 germplasm factors (9GM) from 13 GM, we could even induce almost the entire blastoderm cells to PGCs, which should be a new paradigm for PGC induction and PGC transplantation in teleost fish. The highly efficient iPGCT approach developed by our study is useful not only for editing embryonic lethal genes in a high throughput manner, but also for efficient breeding of aquaculture species by genome editing.

In an effort to provide certain new insights into PGC specification, we conducted additional experiments, including analyzing neonatal mRNA of PGC genes, and added the data related to screening out 9GM from 13GM. The relevant experiment and results are briefly present as below.

First, we conducted additional experiment by visualizing neonatal germplasm mRNA the transplanted iPGCs, in order to clarify whether the iPGCs can activate endogenous PGC

genes and when they initiate their expression. The neonatal mRNA of *tdrd7a* and *ddx4* was detected by single molecule in situ hybridization (smFISH) technology. To distinguish the endogenously transcribed mRNA (neonatal mRNA containing endogenous 3'UTR) from the exogenously loaded mRNA (without endogenous 3'UTR), we designed probes in the UTR regions of mRNA (en_probes) and CDS (ex_probes), respectively (Revised manuscript Supplementary Fig. 1h, i, also see Response New Fig. 7a, b). ex_probes signals of *tdrd7a* and *ddx4* were detected from 30% epiboly to 5-somite stage, while en_probes signals were detected only at 5-somite stage and 1 dpf, and almost all iPGCs could activate endogenous PGC genes (Revised manuscript Supplementary Fig. 1h, i and Fig. 1j, also see Response New Fig. 7a-c). This indicates that iPGC has initiated the expression of germline genes at 5-somite stage. All these data suggest that iPGCs induced by 9GM could activate endogenous PGC genes from 5-somite stage, and we have added these results in the revised manuscript (P7, L119-130).

Response New Figure 7: GFP positive iPGCs activate endogenous PGC genes.

(a) Probes were designed at different positions of mRNA, and single molecule in situ hybridization (smFISH) was used to detect neonatal germline mRNA. Probes for detecting overexpressed mRNA (ex_probes) and probes for neonatal mRNA (en_probes) were designed in the CDS and UTR regions of the mRNA, respectively. ex_probe signals of *tdrd7a* and *ddx4* were detected at 30% epiboly, 75% epiboly and 5-somite stages, while

en_probe signals were detected only at 5-somite stage. (c) Neonatal mRNAs of *ddx4* and *tdrd7a* mRNAs could be clearly detected in the iPGCs at 1dpf.

Second, in previous studies, although Buc could recruit existing nearby germplasm factors to induce ectopic PGCs, the core factors that are sufficient for inducing PGCs are not well defined. In this manuscript, we screened out 9GM from 13GM, and the localized injection experiment strongly suggest that the 9GM are sufficient to induce any blastomere into PGC (Revised manuscript Fig. 1b, g, h, Supplementary Fig. 1b and revised manuscript Fig. 3a, b, also see Response New Fig. 8, Fig. 9a-f). We initially used 13GM, including 9GM related to PGC specialization (*buc*, *piwil1*, *nanos3*, *ddx4*, *dazl*, *dnd*, *tdrd6*, *tdrd7*, *dazap2*) and 4GM related to PGC migration (*rgs14a*, *cxcr4a*, *cxcr4b*, *ca15b*) (Response New Fig. 8) for PGC induction experiment, and we screened out 9GM from 13GM. We have added the relevant data into the revised manuscript Fig. 1b (13GM) and Supplementary Fig. 1b (see also Response new Fig. 9a-d). When we removed the 4GM related to PGC migration, the remaining 9GM were still able to induce PGC efficiently, indicating that the 4GM related to PGC migration do not belong to the key germplasms for PGC induction. These results have been added in the revised manuscript (P5, L73-83, L94-99; P9, L169-178).

Response New Figure 8: An initial 13GM used for PGC induction in this study (Adapted from Agueo et al, 2017; Marlow et al, 2015; Paksa & Raz, 2015)

The gene marked red is component 9GM, and the gene marked blue is the other four genes in 13GM.

Response New Figure 9: Identification of 9GM from 13GM for PGC induction in zebrafish.

(a) At the 1-cell stage, zebrafish embryos were injected with 13GM (50 pg per fraction) or 9GM (50 pg per fraction) to induce iPGCs. GFP-UTRnanos3 was used to label PGCs. The embryos were photographed at 90% epibody. (b) iPGC migrates to the genital ridge of the recipient embryo. (c) iPGCs migrated to the genital ridge of the recipient embryo. (d) iPGCT could significantly improve the success rate of PGC transplantation. Error bars, s.d.; significance values * $P < 0.05$, ** $P < 0.01$, *** $P < 0.001$. (e and f) iPGCs were induced by injecting different combinations of induction factors into a clone at the animal pole or margin of the 128-cell embryos. GFP-UTRnanos3 was used to label iPGCs, and mCherryCAAX was used to label positive cells.

Third, in our ongoing project, we continued to optimize the PGC induction system. 9GM was removed one by one, and we finally identified a combination of germplasm (nGM) containing less factors, which was capable of efficient PGC induction (Response New Fig. 10a-f, Figure for Reviewer 2 only). Since all the experiments in the current manuscript have been performed with 9GM, we only presented the data related to 9GM in this manuscript. The underlying mechanism of nGM inducing PGC is still under careful investigation, thus this piece of work will be presented in a future manuscript.

[Redacted]

Response New Figure 10: Identification of nGM (n<9) from 9GM for PGC induction in zebrafish (Figure for Reviewer 2 only).

(a) PGC was induced by injection of germplasm factors at 1-cell stage. GFP-UTRnanos3 was the marker of PGC. A single factor in 9GM was removed one by one. (b) After the induction factors were removed one by one, the k germplasm factors (kGM) with the greatest influence on iPGC induction were combined to perform iPGC induction. Finally,

the combination of key germlasm for PGC induction was obtained (nGM). (c) The schematic diagram for iPGCT. (d) nGM could induce PGC efficiently, and iPGC migrated to the recipient embryonic genital ridge correctly. (e) A schematic of nGM mRNA injected individually or together in a single cell at the 128-cell stage. (f) At 128-cell stage, nGM mRNA were injected into a single cell alone or together. The migration of iPGC was observed at 5-somite and 1 dpf stage.

Specific comments.

1. The basis for choosing the 9 components is not clear. Are all of them needed? are the RNA concentrations optimized in any way? For a new method, this information would be required.

Question 1: The basis for choosing the 9 components is not clear. Are all of them needed?

Response 12-1: Thank you for pointing out this issue. In the revised manuscript, we added the data related to screening out 9GM from an original combination of 13GM. We initially used 13GM, including 9GM related to PGC specialization (*buc*, *piwil1*, *nanos3*, *ddx4*, *dazl*, *dnd*, *tdrd6*, *tdrd7*, *dazap2*) and 4GM related to PGC migration (*rgs14a*, *cxcr4a*, *cxcr4b*, *ca15b*) (Response New Fig. 8) for PGC induction experiment, and we screened out 9GM from 13GM. We have added the relevant data into the revised manuscript Fig. 1b (13GM) and Supplementary Fig. 1b (also see Response New Fig. 9a, b). When we removed the 4GM related to PGC migration, the remaining 9GM were still able to induce PGC efficiently (Revised manuscript Fig. 1g, h, also see Response New Figure 9c, d), indicating that the 4GM related to PGC migration do not belong to the key germlasms for PGC induction. These results have been added in the revised manuscript (P5, L73-83, L94-99).

In our ongoing project, we continued to optimize the PGC induction system. 9GM was removed one by one, and finally identified a combination of germlasms (nGM) containing less factors, which was capable of efficient PGC induction (Response New Figure 10a-f, Figure for Reviewer 2 only). Since all the experiments in the current manuscript have been performed with 9GM, we only presented the data related to 9GM in this manuscript. The underlying mechanism of nGM inducing PGC is still under careful investigation, thus this piece of work will be presented in a future manuscript.

Question 2: Are the RNA concentrations optimized in any way?

Response 12-2: Thank you for pointing out this issue. According to your advice, we added the relevant results that were missing from the original manuscript (P5, L83-88). Initially, we have optimized the RNA concentration. At low dose (25 pg per factor), GFP-UTR*nanos3* was weakly positive. At high doses (100 pg per factor), the proliferation of embryonic cells was strongly inhibited. In contrast, iPGC can be effectively induced only at moderate dose (50 pg per factor) (Revised manuscript Supplementary Fig. 1a, also see Response New Fig. 11). Therefore, PGC induction was performed with moderate dose of germlasm factors in subsequent experiments.

Response New Figure 11: Injection of high dose (100 pg for each factor) of 9GM strongly inhibited cell proliferation, and injection of low dose (25 pg for each factor) of 9GM did not induce GFP-UTR*nanos3* positive cells with high efficiency. In contrast, injection of moderate dose (50 pg for each factor) of 9GM could effectively induce iPGCs.

2. What is special about the iPGCs that does not support the development of female fish? The possible explanation provided by the authors "In iPGCT embryos, apoptosis or immune rejection of donor iPGCs can lead to a stronger immune response and eventually female-to-male reversal. " It is not clear how intraspecies transplantation (the animals are inbred) leads to rejection. Why only of oocytes.

Response 13: Thank you for pointing out this issue. To explore the reasons why the iPGCs that does not support the development of female fish, we continuously tracked the proliferation and differentiation of iPGCs (Revised manuscript Supplementary Fig. 2a-d, also see Response New Fig. 12a-d).

From 12 dpf to 32 dpf, the number of germ cells in iPGCT fish was significantly lower than that in control zebrafish, and the developmental progress of germ cells in iPGCT fish was also obviously retarded when compared with the control (Revised manuscript Supplementary Fig. 2c, also see Response New Fig. 12c). The lower number of germ cells in juvenile fish might lead to final masculinization of the iPGCT zebrafish, since a sufficient number of germ cells is required for female development in zebrafish.

We also observed that the iPGCs in a few iPGCT embryos could still develop into stage I oocytes at 50 dpf (Revised manuscript Supplementary Fig. 2d, also see Response New Fig. 12d), but we do not know whether these transplants could finally develop into females.

We have added the relevant data and revised the Discussion accordingly (Revised manuscript Supplementary Fig. 2a-d, P15, L295-309).

Response New Figure 12: The process of iPGC proliferation and differentiation.

(a-d) Immunofluorescence imaging was used to track the process of iPGCs proliferation and differentiation. The enlarged image in Supplementary Figure 2d showed the chromatin status in the nucleus of the arrowheads. The iPGCT gonads of 40dpf and 50dpf showed the ovariform gonad, while the testicular form gonad was shown in the upper right corner of the image.

3. What is the fate of the progeny of animals that were obtained from the iPGCs? This point is important if the method is used for the generation of adult lines carrying transgenes and mutations. Similarly, using the procedure for maintaining wild-type fish for species preservation and agriculture would require the generation of healthy, fertile, and normal adults.

Response 14: Thank you for your suggestion! The progeny of animals obtained from the iPGCs were able to grow and reproduce normally, and we have added the data into the revised manuscript (Revised manuscript Fig. 2g, Fig. 4i, also see Response New Fig. 13a, b).

Response new Figure 13: The progeny of animals that were obtained from the iPGCs are able to grow and reproduce normally.

F1 generations obtained through the iPGCT continue to breed, and obtained the progeny (F2) that grew normally.

4. The methods section and the figure legends are very minimal and provide very little information regarding the precise experimental design and details. Similarly, the movies and tables are not accompanied by legends. What is the context of the green cells in Fig 1e, 10d embryo. There are also multiple mistakes. For example, in Fig 2c itself the stage is 32dpf, while the legend states that the stage is 25dpf. Also, in Figure 2d, it is not clear from the panel what the difference between normal testes (Tg(cmv:GFP)) and the dnd MO. What was the protocol used for the qRT PCR? Another example is in Fig 2m which has 12 panels with a minimal description. 'Characterization of the recombinant gonads by immunofluorescence'...what is 1 and what is 2. Magnification?

Response 15-1: Sorry for these mistakes. We have corrected these mistakes and added detailed figure legends and movies legends.

What is the context of the green cells in Fig 1e, 10d embryo.

Response 15-2: Thank you for pointing out this issue. The green cells were labeled with GFP-UTR*nanos3* and were derived from the donor iPGCs. We have added detailed figure legends in revised manuscript (Revised manuscript Fig. 1i and P26, L646-648).

in Figure 2d, it is not clear from the panel what the difference between normal testes (Tg(cmv:GFP)) and the dnd MO.

Response 15-3: Thank you for pointing out this issue. We have added anatomical figures of the gonads in order to see the differences between the different groups more clearly (Revised manuscript Supplementary Fig. 2e, also see Response New Fig. 14).

Response New Figure 14: The anatomical figure of the gonads.

The gonadal differences were more clearly observed by stripping the gonads from different groups.

What was the protocol used for the qRT PCR?

Response 15-4: Sorry for missing this protocol. We have added the details in the figure legends and methods in the revised manuscript (P19, L387-395). Briefly, relative

expression level (ΔCt) of each gene in every sample was obtained by qRT-PCR according to previous studies (Livak and Schmittgen, *Methods*, 2001, doi: 10.1006/meth.2001.1262), and presented as a heatmap according to previous reports (Mao, et al., *Stem cell reports*, 2016, doi: 10.1016/j.stemcr.2016.09.002; Rahman, et al., *Small*, 2020, doi: 10.1002/smll.202000272).

5. It would recommend following other germ-cell characteristics such as germ-cell granules. Are all the cells shown in Fig 1B show perinuclear granules?

Response 16: Thank you for pointing out this issue. According to your suggestion, we followed other germ-cell characteristics, such as germ-cell granules, and transcription of neonatal germplasm mRNA to characterize the iPGC, and added the new data as follows.

First, at shield stage and 8 dpf, we found that Vasa protein was granulated in the perinuclear region of iPGCs (Revised manuscript Supplementary Fig. 1e, Fig. 1i, also see Response New Fig. 15a, b). In addition, *tdrd7a* and *ddx4* mRNA showed perinuclear granule-specific distribution in almost all iPGCs at 5-somites and 1 dpf stages (Revised manuscript Fig. 1j, Supplementary Fig. 1i, also see Response New Fig. 15c, d).

Second, to identify neonatal mRNA containing endogenous 3'UTR from the exogenously loaded mRNA without endogenous 3'UTR, we designed probes in the UTR regions of mRNA (en_probes) and CDS (ex_probes), respectively, and detected them by single molecule in situ hybridization (smFISH) technology. ex_probes signals of *tdrd7a* and *ddx4* were detected from 30% epiboly to 5-somite stage, while en_probes signals were detected only at 5-somite stage and 1 dpf, and almost all iPGCs could activate endogenous PGC genes (Revised manuscript Supplementary Fig. 1h, i and Fig. 1j, also see Response new Fig. 15c-e). This indicated that iPGC has initiated the expression of germplasm genes at 5-somite stage and possessed germ-cell characteristics.

We have added these results in the revised manuscript (P6-7, L114-130).

Response new Figure 15: Neonatal mRNA can be detected in the iPGC.

(a and b) At shield stage and 8 dpf, GFP-UTR*nanos3* positive cells showed Vasa antibody positive in embryos overexpressing germplasm factors. (c-e) Probes were designed at different positions of mRNA, and single molecule in situ hybridization (smFISH) was used to detect neonatal mRNA. Probes for detecting overexpressed mRNA (ex_probes) and probes for neonatal mRNA (en_probes) were designed in the CDS and UTR regions of the mRNA, respectively. en_probes and ex_probes signals of *tdrd7a* and *ddx4* were detected at 5-somite stage and 1 dpf.

6. The statement that "In contrast to ePGCs, which generally divided slowly, blastula cell-derived iPGCs still divided rapidly in host embryos, mimicking early embryonic cells (Movie 2)" is not observed in the movie and there is no quantitation of the division rate of the iPGCs vs the ePGCs.

Response 17: Thank you for pointing out this issue. According to your suggestion, we have done additional experiment, and added quantitation of the division rate of the iPGCs vs the ePGCs in revised manuscript Fig.1e, f, Supplementary Fig 1c, Supplementary Movie 2, also see Response New Fig. 16a-c.

At 30% epiboly stage, around 10 iPGCs were transplanted to the recipient embryo. At 10-somites stage, the number of iPGC reaches around 80, which is much higher than the number of ePGC (an average of 20) (Revised manuscript Fig.1e, f, Supplementary Fig. 1c, Supplementary Movie 2, also see Response New Fig. 16a-c). The fast division of iPGCs was presented as a time-lapse panel in revised manuscript Supplementary Fig. 1c (also see Response New Fig. 16c). This indicates that iPGC may have the characteristics of rapid proliferation similar to somatic cells in the early stage, and gradually change to PGC characteristics in the somite stage. This is consistent with the expression of endogenous germplasm factors at 5-somites stage (Revised manuscript Fig.1j, Supplementary Fig 1h, i, also see Response New Fig. 15c-e). We have added results to the revised manuscript (P6, L99-110).

Response new Figure 16: iPGC can proliferate rapidly.

(a) GFP-UTR*nanos3* was used to label PGC and ePGC. Since GFP-UTR*nanos3* could not label endogenous PGC at 30% epiboly stage, Piwil1 antibody was used to label endogenous PGCs. (b) The number of PGCs in wild type and iPGCT at different stages. * $P < 0.05$, ** $P < 0.01$, *** $P < 0.001$. (c) About 10 iPGCs were transplanted into *Tg(piwil1:egfp-UTRnanos3)* receptors, the endogenous PGCS (ePGC) were labeled green (White arrow marked), and the iPGCs were labeled by mCherry-UTR*nanos3*. iPGC and ePGC proliferation were tracked after transplantation.

7. Movie 4 could be shortened and the efficiency it should show is not obvious from the presentation.

Response 18: Thank you! We've shortened the movie (Revised manuscript Supplementary Movie 3).

This video mainly shows the iPCT operation process, in which a large number of iPCTs could be aspirated at one time, and 10-20 cells were transplanted to each host embryo. Therefore, one donor embryo could provide iPCT for at least 30 host embryos, highlighting the characteristics of high throughput by iPCT.

8. The actual phenotypes induced by CRISPR should be presented.

Response 19: Thank you for pointing out this issue. We have added the relevant data into the Revised manuscript Supplementary Fig. 3b-d, also see Response New Fig. 17a-c.

Response new Figure 17: The phenotypes of progeny obtained by combining blastomere genome editing with PGC induction.

(a) The F1 generation obtained from chimeras (bmp7a #1) produced severely developmentally defective embryos. (b) The F2 generation obtained from chimeras (pou5f3 #2) produced severely. (c) The F1 generation obtained from chimeras (tyr #2) produced embryos without pigment.

Reviewer #3:

The manuscript by Wang et al. describes the novel strategy, using induced primordial germ cells (iPGCs) in vivo, to generate knocking-out (KO) or knocking-in (KI) zebrafish. The authors demonstrate that ectopic expression of 9 germplasm factors (9GM) in the early zebrafish embryo is sufficient to make somatic cells to convert into the germ cell lineage, thereby making iPGCs in vivo. In combination of authentic genome editing strategies, transplantation of iPGCs in wild-type host embryos allows for efficient generation of KO or KI zebrafish. Furthermore, using the same strategy, zebrafish donor iPGCs can be used for genetic manipulations of another teleost fish, *Gobiocypris rarus* (Gr).

Generation of iPGCs in zebrafish in vivo is novel, and is highly applicable for in vivo genome editing, in particular for genes that are required for early development. Thus, this work can lead to a paradigm shift for genome editing in zebrafish. I would support this work for publication in Nature Communications if my concerns were clarified.

Thank you very much for your appreciation of our work.

(Major points)

1. Figure 3f require more experiments to increase the number of the success cases. Just showing one case each for *pou5f3* or *tyr* with low efficiency (10%) is not convincing.

Response 20: Thank you for pointing out this issue. We have conducted additional experiments to show the high efficiency of iPGC-targeted genome-editing by completely removing endogenous PGCs. The new data was added in Revised manuscript Fig. 3g, h (also see Response New Fig. 18a, b), and revised main text (P10, L196-202).

Briefly, in order to remove endogenous PGCs, we now injected *dnd* MO2 (ATGTCTCCGACCATCTGTGATGATG, Gross-Thebing et al., Developmental Cell, 2017, doi: 10.1016/j.devcel.2017.11.019), which could inhibit the translation of endogenous *dnd* mRNA, but not exogenous *dnd* mRNA. In this experiment, the efficiency of gametic mutation is greatly improved.

Response new Figure 18: Genome editing and PGC induction were combined to efficiently knock out endogenous genes.

(a) Schematic representation of simultaneous genome editing and PGC induction in any single blastomere at the 128-cell stage. Embryos were injected with a low dose of dnd MO2 (10 μ M) at 1-cell stage to eliminate the endogenous PGC. iPGCs migrated to the genital ridge and eventually produced genome-edited gametes. (b) Mutation efficiencies of gametes. A total of 10 embryos obtained from hybridization of chimeras and wild-type were used to calculate gamete mutation efficiency.

2. The detailed design and procedure of the KI strategy should be shown in supplemental Figure and materials & methods, including how long homologous sequences is used on either the 5'- or 3'- region and all the sequences of gRNAs (sox19b and nanog). It seems that 'NHEJ' is used for nanog knock-in as well as 'MMEJ' for mpx and sox19b. This should be described in materials & methods, as well.

Response 21: Thank you for pointing out this issue. We have added detailed explanations in the revised manuscript methods (P18, L363-374) and Supplementary Table 3.

(Minor points)

Line 68, Reference 11 is not a proper article to cite. Instead, here, the authors should cite the original paper for each of 9 germlasm factors separately.

Response 22: Thank you for pointing out this issue. We have cited the original paper for each of 9 germlasm factors separately (P5, L75-76).

Line 101. , 'asymmetric migration of iPGCs' is unclear.

Response 23: Thank you for pointing out this issue. "asymmetric migration of iPGCs" mean that iPGC migrated mainly to the genital ridge on one side of the recipient embryo. We have added a clearer explanation.

Figure 2m needs proper labeling of '1'.

Response 24: We have corrected it (see Fig. 2f).

Line 189, 'As shown in Figure 6e,...' but I cannot find the figure.

Response 25: Sorry for this mistake. It should be Figure 5e. We have corrected it.

Line 412, Figure legend1 (b), it should be ...buc or 9GM mRNA to induce PGCs, instead of buc and 9GM mRNA.

Response 26: Sorry for this mistake. We have corrected it (P26, L642-643).

Line 414, Spell induced PGC transplantation (iPGCT), instead of PGCT.

Response 27: Sorry for this mistake. We have corrected it.

Line 416. The legend for (e) requires how to visualize the PGC-specific protein Vasa (immunohistochemistry or GFP reporter?).

Response 28: Thank you for pointing out this issue. According to your suggestion, we stained the iPGC with Vasa antibody and showed GFP-UTR*nanos3* at the same time.

It was found that GFP positive cells showed Vasa antibody positive (Revised manuscript Fig. 1i, also see Response New Fig. 19a), suggesting that iPGC expressed PGC-specific marker. In addition, GFP-UTR*nanos3*-positive ePGC was not detected after injecting *dnd* MO at 1 dpf embryos. In addition, Vasa antibody was not detected in *dnd* MO embryos at 8 dpf (Revised manuscript Supplementary Fig. 1f, g, also see Response New Fig. 19b, c). These results indicate that endogenous PGC is completely removed. We have added results to the revised manuscript (P6-7, L114-119).

Response new Figure 19: (a) Immunofluorescence staining was used to label the PGC-specific protein Vasa at 8 dpf. iPGC was labeled with GFP-UTR*nanos3*. (b) At 1 dpf GFP-UTR*nanos3* was not detectable in embryos injected with *dnd* MO. (c). At 8dpf, Vasa antibodies were not detectable in embryos injected with *dnd* MO.

Line 417, p-values need to be shown or in the corresponding main text.

Response 29: We have added p-values to the revised manuscript.

Line 427, '... meiosis and meiosis...' should be '...meiosis and mitosis...'.

Response 30: Thank you! We have corrected it (P28, L665-666).

Line 434, the p-values need to be shown or in the corresponding main text.

Response 31: Thank you! We have added p-values to the manuscript (P26-27, L649-652).

Line 473, 'MMEJ' needs to be spelled in full.

Response 32: We have added the full spelling of MMEJ (microhomology-mediated end-joining) (P12-13, L245-247).

REVIEWERS' COMMENTS

Reviewer #1 (Remarks to the Author):

The authors have appropriately addressed all my concerns by adding new data and descriptions.

Reviewer #2 (Remarks to the Author):

The revised version of the manuscript "Induction of primordial germ cells from fish embryonic cells by nine factors" by Wang et al is improved as compared with the original version.

The authors addressed most of the points I raised and in my opinion the paper can prove to be an important technical advance in basic research and aquaculture.

The remaining issues are listed below. In short, the impact of the paper would be much higher if the authors ask a native English speaker to comment on the language. I am listing a few examples as specific points, but some sections of the manuscript are difficult to understand just due to language / wording. This point should not be difficult to address. A second issue relates to the title of the paper, which I believe does not follow the current common terminology for specification of germ cells. The current title will also become obsolete soon, since according to the authors they already know that not all nine factors are essential. Thus, as soon as for example 8 factors prove to be sufficient, the current title loses its value. I suggested an alternative in point 1. Last, the references cited are very poorly chosen. I highlighted and explained below a few examples (in particular see point 6). This issue should also be relatively easy to address.

1.

Two issues about the title –

-In recent literature, in the context of germ-cell development, "Induction" primarily refers to the effect of externally-produced/supplied factors on cells that become germ cells, as in the mouse for example. In animals whose germ cells are specified by inheritance of maternally-provided factors, the process is usually termed "preformation". The authors present these definitions in the introduction, but then do not use them, which is confusing.

-The most successful experiments, which are likely to represent a procedure that will be mostly used in the future, were conducted in zebrafish. People interested in other fish species or in *Gobiocypris rarus* would be attracted to reading the paper by having "Gobiocypris rarus" as a keyword and by mentioning it in the abstract (as is done). Using the word "fish", might be interpreted as if the findings are relevant for all fish species. This was not tested and, in some species, the presence of germ plasm was not even shown (e.g. basal species).

I would therefore change the title of the paper into - "Efficient conversion of zebrafish blastomeres into primordial germ cells by overexpression of germplasm components". See next point for why not using the "nine factors" in the title.

2. Since in their response the authors show that not all 9 factors are needed, it is a pity that the term "9GM" is used, since it will become irrelevant in the very near future and will disappear. It is especially not good to have it in the title. When introducing the converting RNAs I would define the mixture as something like "PGC-converting mix – PCM" and state that it is likely that less than 9 factors are needed and that it is a point for future investigation. This is mentioned at the end of the discussion, but given that a result is already there, better

to mention it briefly earlier.

3. For the same reason as for the title (point 1), I would not use the term “induced PGCs (iPGCs)” throughout the text, but rather something like “PGCs produced by PCM injection”.

4. “To our knowledge, this is the first success in inducing PGC in vivo using preformation theory, which creates a new paradigm for PGC induction and PGCT in teleost fish.” Change to “ Together, we present a novel and efficient method for generating PGCs in a teleost, manipulating them and reintroducing them into host embryos, a technique that will have a strong impact in basic research and aquaculture”.

5. “...designed to increase the efficiency of genetic manipulation..”. Add an “s” after “manipulation”. In general, the paper would very very strongly benefit from English editing, beyond the few examples I mention.

6. References – At many places the authors do not cite the correct source, a point that should be thoroughly addressed. For example – “ Maternally inherited germplasm factors, such as vasa 15, dazl 16, piwil1 17, dnd 18, nanos3 19, tdrd 20, 21, etc., are deposited in presumptive primordial germ cells (pPGCs), leading them to segregate from the somatic lineage and eventually form PGCs.”. First, “etc.” is used several times instead of “for example” or “such as”. In the sentence above, it is not clear that the authors refer to fish only (as the references reveal). Most of the factors mentioned were first found in other organisms. If the statement is general, the corresponding papers should be presented and those in which the molecule was first described. If the authors want to refer only to zebrafish, they should state that and again, cite the first time it was identified as a germ-cell marker in fish. e.g for vasa (a paper from 2014 is cited twice – 15 and 31), instead of DOI:

10.1242/dev.124.16.3157 (a 1997 manuscript). If needed, when referring to actual molecular / cellular function, more recent papers can be cited as well. In contrast, generally, review articles should also be the most recent ones. Overall, the choice of papers to cite is very poor. e.g. reference 25 is not the best one for showing that zebrafish is an important model.

7. “ While the induction of PGC-like cells from embryonic stem cells or pluripotent stem cells has been achieved in mice 22, 23, rats 6 and northern white rhinoceros 24, and the induced PGCs (iPGCs) could be further differentiated into functional gametes after transplantation into germ cell-deficient host animals 6, 22, 23, 24.” - “While” should not be used at the beginning.

8. “Nevertheless, these induction strategies are all based on the epigenesis model, and there are no successful reports of PGC induction using the preformation model in oviparous species.” As mentioned above, better not to use the “induction” in the “preformation” context. In any case, increasing the number of germ cells was described before for example in zebrafish (Buc) and Drosophila (DOI: 10.1038/358387a0 , (where “induction” is used, before the mouse mechanisms were analysed thoroughly)).

9. In Fig1a – do not use “etc”.

10. “In zebrafish, Drosophila and other animals, PGCs are specialized at different sites in the embryo and must migrate to the site of gonadal development for proliferation and differentiation 36, 38.” The PGCs are “specified”, not “specialized”. What does “different sites” mean?

11. Cxcr4a is mentioned as one of the 4 proteins important for migration. This is not the case and it is only Cxcr4b (which is not cited - DOI: 10.1016/s0092-8674(02)01135-2)

12. “...further confirming that the germ cells in the host embryos were from the transplanted iPGCs”. Use “were derived” instead or “were”.

13. In Fig 3a, what is the explanation for the fact that control and buc-injected cells do not reach the gonad? If the 9 factors (without the “migration proteins”) allow cells to get to the target, what do the control transplanted cells miss? Could state that the 9-factor mix most likely lead to expression of the motility module.

14. In supplemental Fig3, the F2 phenotypes are presumably a result of incrosses – should

be indicated in the legend. It is not clear how one obtains the phenotype in F1 (panel d). Is the BMP mutation dominant? please explain, and check if references 42 and 43 are indeed relevant here.

15. "When Gr_iPGCs were transplanted into ePGC-depleted zebrafish embryos (Gr_iPGCT_Dr), these Gr_iPGCs were able to migrate correctly to the zebrafish genital ridges with a rate of almost 100%, which was much higher than the rate of traditional PGCT (Gr_PGCT_Dr) (Fig. 4c-e and Supplementary Movie 6)." - what one understands from this sentence is that the migration of the Gr_iPGC is more CORRECT than that of "traditional PGCT". This is not shown in the figures/movie. Rather, the differences in colonization of the gonad are likely to reflect the effect of the larger initial number of cells.

16. "This suggests that the developmental process of germ cells may be faster in Gr_iPGCT_Dr than in Gr 46." Could this reflect differences in temperature, rather than "faster developmental process"? How does the finding the "the other type expressed Vasa as strongly as Gr, and more than 10 germ cells were grouped together (Fig. 4g)" suggests faster development?

17. "It may be due to the fact that germplasm factors derived from zebrafish are used, leading to insufficient...." " Do the authors mean "used up"?

18. "...assume that the *in vivo* induction of iPGCs using..." – *in vivo* should be in italics.

19. The "iPGCT greatly improved the knock-in efficiency of gametes" section fits better before the "Xenogametes generated by iPGCT" part.

20. "In zebrafish and medaka fish, previous studies have focused on PGC induction via single germplasm factors and failed in PGC induction from the essence of PGC formation 33, 59". The use of the word "essence" is not clear here. In addition, Buc over expression was shown to lead to an increase in germ-cell number, albeit not as efficiently as presented in this manuscript.

21. "Midas touch" – better to use a more scientific term.

Reviewer #3 (Remarks to the Author):

The revised manuscript by Wang et al. has been improved to a satisfactory degree. Now I recommend this study for publication in Nature Communications.

(Minor point)

Figure 3h needs to explain what #1-#5 are in the legend.

Revision summary and Point-by-Point Response to Reviewers

1. Summary of major comments from the reviewers.

Based on the requests and comments from the reviewers, we have performed further analyses and additional experiments. Our major revision and new supporting data are summarized and listed in the following Table 1.

Table 1. Revision summary and new supporting data to address major comments from the reviewers.

	Questions	Reviewers	Clarification on the original submission data and New data supporting
1	The remaining issues are listed below. In short, the impact of the paper would be much higher if the authors ask a native English speaker to comment on the language. I am listing a few examples as specific points, but some sections of the manuscript are difficult to understand just due to language / wording. This point should not be difficult to address. A second issue relates to the title of the paper, which I believe does not follow the current common terminology for specification of germ cells. The current title will also become obsolete soon, since according to the authors they already know that not all nine factors are essential. Thus, as soon as for example 8 factors prove to be sufficient, the current	2#: Q1	Thank you very much for your appreciation and critical advices. We have asked a native English speaker to revise the language of the manuscript. The references in this manuscript are also verified and changed. In addition, according to your suggestion, we have changed the title of the manuscript to "Induced formation of primordial germ cells from zebrafish blastomeres by germlasm factors".

	title losses its value. I suggested an alternative in point 1. Last, the references cited are very poorly chosen. I highlighted and explained below a few examples (in particular see point 6). This issue should also be relatively easy to address.		
2	I would therefore change the title of the paper into - "Efficient conversion of zebrafish blastomeres into primordial germ cells by overexpression of germplasm components". See next point for why not using the "nine factors" in the title.	2#: Q2	Yes, thank you for your advice. 'nine germplasm factor' does not fit well in the title. We have changed to "Induced formation of primordial germ cells from zebrafish blastomeres by germplasm factors". We agree that "Induction" primarily refers to the effect of externally-produced/supplied factors on cells that become germ cells. In fact, the embryonic cells become germ cells by externally-supplied factors "9GMs" in this study, which perfectly reflects the essence of "induction" of PGCs from embryonic cells. Although "preformation" is the mechanism of PGC specification in fish, we show here that a combination of "preformation" factors has PGC induction activity for the non-PGC embryonic cells. Therefore, we believe that "Induction" or "Induced formation" in the revised title can more faithfully express the events of cell fate conversion from embryonic cells to PGCs.
3	Since in their response the authors show that not all 9 factors are needed, it is a pity that the term "9GM" is used, since it will become irrelevant in the very near future and will disappear. It is especially not good to have it in the title. When introducing the converting RNAs I would define the mixture as something like "PGC-converting mix – PCM" and state that it is likely that less than 9 factors are needed and that it is a point for future	2#: Q3	Thank you for the suggestion. However, to keep the current manuscript as attractive as it can, we prefer to keep the term "9GMs" in the manuscript. In fact, when we originally prepared the manuscript, we considered that all these 9GMs were necessary for PGC induction, indicating that the term "9GMs" was scientifically reasonable in the current paper. On the other hand, if we define the current mixture as "PGC-converting mix – PCM", it might be difficult for us to clearly state the difference between the future PCM with the current PCM.

	investigation. This is mentioned at the end of the discussion, but given that a result is already there, better to mention it briefly earlier.		
4	For the same reason as for the title (point 1), I would not use the term “induced PGCs (iPGCs)” throughout the text, but rather something like “PGCs produced by PCM injection”.	2#: Q4	Thank you for your suggestion. However, we prefer to keep the term “iPGCs”. The main reasons have been explained in Response 2.
5	“To our knowledge, this is the first success in inducing PGC in vivo using preformation theory, which creates a new paradigm for PGC induction and PGCT in teleost fish.” Change to “ Together, we present a novel and efficient method for generating PGCs in a teleost, manipulating them and reintroducing them into host embryos, a technique that will have a strong impact in basic research and aquaculture”.	2#: Q5	Thank you for your suggestion. We have corrected it.
6	“...designed to increase the efficiency of genetic manipulation..”. Add an “s” after “manipulation”. In general, the paper would very very strongly benefit from English editing, beyond the few examples I mention.	2#: Q6	Thank you. We have corrected it.
7	References – At many places the authors do not cite the correct source, a point that should be thoroughly addressed. For	2#: Q7	Thank you. We have corrected all these references according to your suggestion.

example – “ Maternally inherited germlasm factors, such as vasa 15, dazl 16, piwil1 17, dnd 18, nanos3 19, tdrd 20, 21, etc., are deposited in presumptive primordial germ cells (pPGCs), leading them to segregate from the somatic lineage and eventually form PGCs.”. First, “etc.” is used several times instead of “for example” or “such as”. In the sentence above, it is not clear that the authors refer to fish only (as the references reveal). Most of the factors mentioned were first found in other organisms. If the statement is general, the corresponding papers should be presented and those in which the molecule was first described. If the authors want to refer only to zebrafish, they should state that and again, cite the first time it was identified as a germ-cell marker in fish. e.g for vasa (a paper from 2014 is cited twice – 15 and 31), instead of DOI: 10.1242/dev.124.16.3157 (a 1997 manuscript). If needed, when referring to actual molecular / cellular function, more recent papers can be cited as well. In contrast, generally, review articles should also be the most recent ones.		
---	--	--

	Overall, the choice of papers to cite is very poor. e.g. reference 25 is not the best one for showing that zebrafish is an important model.		
8	“ While the induction of PGC-like cells from embryonic stem cells or pluripotent stem cells has been achieved in mice 22, 23, rats 6 and northern white rhinoceros 24, and the induced PGCs (iPGCs) could be further differentiated into functional gametes after transplantation into germ cell-deficient host animals 6, 22, 23, 24.” - “While” should not be used at the beginning.	2#: Q8	Thank you. We have corrected it.
9	“ Nevertheless, these induction strategies are all based on the epigenesis model, and there are no successful reports of PGC induction using the preformation model in oviparous species.” As mentioned above, better not to use the “induction” in the “preformation” context. In any case, increasing the number of germ cells was described before for example in zebrafish (Buc) and Drosophila (DOI: 10.1038/358387a0 , (where “induction” is used, before the mouse mechanisms were analysed thoroughly)).	2#: Q9	Thank you. We prefer to keep the term “PGC induction” here. The main reasons have been described in Response 2.
10	In Fig1a – do not use “etc”.	2#: Q10	Thank you. We have corrected it.

11	" In zebrafish, Drosophila and other animals, PGCs are specialized at different sites in the embryo and must migrate to the site of gonadal development for proliferation and differentiation 36, 38." The PGCs are "specified", not "specialized". What does "different sites" mean?	2#: Q11	Thank you. We have corrected it and deleted the phrase "different sites".
12	Cxcr4a is mentioned as one of the 4 proteins important for migration. This is not the case and it is only Cxcr4b (which is not cited - DOI: 10.1016/s0092-8674(02)01135-2)	2#: Q12	Thank you. We have added the reference for Cxcr4a (ref 34).
13	" ... further confirming that the germ cells in the host embryos were from the transplanted iPGCs". Use "were derived" instead or "were".	2#: Q13	Thank you. We have corrected it.
14	In Fig 3a, what is the explanation for the fact that control and buc-injected cells do not reach the gonad? If the 9 factors (without the "migration proteins") allow cells to get to the target, what do the control transplanted cells miss? Could state that the 9-factor mix most likely lead to expression of the motility module.	2#: Q14	Thank you for raising the question. In fact, the control and buc-injected cells located in the animal pole, which endogenously lack the PGC mobility module. In contrast, the 9GM-injected cells were converted to PGCs, strongly suggesting that 9GMs overexpression led to expression of the mobility module. We have briefly discussed this point in the revision.
15	In supplemental Fig3, the F2 phenotypes are presumably a result of incrosses – should be indicated in the legend. It is not clear how one obtains	2#: Q15	Thank you for pointing out this issue. We have added detail explanation in the legend. The F1 generations were obtained by crossing chimera with wild type. The F2 embryos derived from F1 generations through incrossing showed obvious developmental defects (Revised manuscript Supplementary Fig. 3d, e, f).

	the phenotype in F1 (panel d). Is the BMP mutation dominant? please explain, and check if references 42 and 43 are indeed relevant here.		
16	“ When Gr_iPGCs were transplanted into ePGC-depleted zebrafish embryos (Gr_iPGCT_Dr), these Gr_iPGCs were able to migrate correctly to the zebrafish genital ridges with a rate of almost 100%, which was much higher than the rate of traditional PGCT (Gr_PGCT_Dr) (Fig. 4c-e and Supplementary Movie 6).”- what one understands from this sentence is that the migration of the Gr_iPGC is more CORRECT than that of “traditional PGCT”. This is not shown in the figures/movie. Rather, the differences in colonization of the gonad are likely to reflect the effect of the larger initial number of cells.	2#: Q16	Sorry for the lack of clarity in the sentence. We have changed the sentence of “which was much higher than the rate of traditional PGCT (Gr_PGCT_Dr)” to “The percentage of embryos with PGCs in genital ridge using iPGCT was much higher than that using traditional PGCT” (P11, L218-220).
17	“ This suggests that the developmental process of germ cells may be faster in Gr_iPGCT_Dr than in Gr 46.” Could this reflect differences in temperature, rather than “faster developmental process”? How does the finding the “the other type expressed Vasa as strongly as Gr, and more than 10 germ cells were grouped together	2#: Q17	Thank you for pointing out this issue. Temperature is indeed one of the important factors affecting the rate of gonad development. We have deleted this sentence in the revision.

	(Fig. 4g)” suggests faster development?		
18	“It may be due to the fact that germlasm factors derived from zebrafish are used, leading to insufficient.... “ Do the authors mean “used up”?	2#: Q18	Thank you! We have changed this sentence to “because germlasm factors derived from zebrafish were utilized”.
19	“...assume that the in vivo induction of iPGCs using....” – in vivo should be in italics.	2#: Q19	Thank you. We have corrected it.
20	The “iPGCT greatly improved the knock-in efficiency of gametes” section fits better before the “Xenogametes generated by iPGCT” part.	2#: Q20	Thank you for the suggestion! We prefer to keep the “Xenogametes generated by iPGCT” section before the “iPGCT greatly improved the knock-in efficiency of gametes” section, since we did not tried genome editing or gene knock-in in xenograft experiment. We just want to briefly show that the iPGCT approach could be expanded to another fish species.
21	“In zebrafish and medaka fish, previous studies have focused on PGC induction via single germlasm factors and failed in PGC induction from the essence of PGC formation 33, 59”. The use of the word “essence” is not clear here. In addition, Buc over expression was shown to lead to an increase in germ-cell number, albeit not as efficiently as presented in this manuscript.	2#: Q21	Thank you for pointing out this issue. We have changed the word “essence” to “mechanism”.
22	“Midas touch” – better to use a more scientific term.	2#: Q22	Thank you! We have deleted this phrase here.
23	Figure 3h needs to explain what #1-#5 are in the legend.	3#: Q1	Thank you for pointing out this issue. #1-#5 refers to the mutation efficiency of 5 F0 fish (#1-#5) gametes. We have added an explanation to the legend (P36-37 , L727-729).

2. Point-by-point responses to reviewers

Reviewer #2 (Remarks to the Author):

The revised version of the manuscript “Induction of primordial germ cells from fish embryonic cells by nine factors” by Wang et al is improved as compared with the original version.

The authors addressed most of the points I raised and in my opinion the paper can prove to be an important technical advance in basic research and aquaculture.

The remaining issues are listed below. In short, the impact of the paper would be much higher if the authors ask a native English speaker to comment on the language. I am listing a few examples as specific points, but some sections of the manuscript are difficult to understand just due to language / wording. This point should not be difficult to address. A second issue relates to the title of the paper, which I believe does not follow the current common terminology for specification of germ cells. The current title will also become obsolete soon, since according to the authors they already know that not all nine factors are essential. Thus, as soon as for example 8 factors prove to be sufficient, the current title loses its value. I suggested an alternative in point 1. Last, the references cited are very poorly chosen. I highlighted and explained below a few examples (in particular see point 6). This issue should also be relatively easy to address.

Response 1: Thank you very much for your appreciation and critical advices. We have asked a native English speaker to revise the language of the manuscript. The references in this manuscript are also verified and changed. In addition, according to your suggestion, we have changed the title of the manuscript to "Induced formation of primordial germ cells from zebrafish blastomeres by germplasm factors".

1.

Two issues about the title –

-In recent literature, in the context of germ-cell development, “Induction” primarily refers to the effect of externally-produced/supplied factors on cells that become germ cells, as in the mouse for example. In animals whose germ cells are specified by inheritance of maternally-provided factors, the process is usually termed “preformation”. The authors present these definitions in the introduction, but then do not use them, which is confusing.

-The most successful experiments, which are likely to represent a procedure that will be mostly used in the future, were conducted in zebrafish. People interested in other fish species or in *Gobiocypris rarus* would be attracted to reading the paper by having “*Gobiocypris rarus*” as a keyword and by mentioning it in the abstract (as is done). Using the word “fish”, might be interpreted as if the findings are relevant for all fish species. This was not tested and, in some species, the presence of germ plasm was not even shown (e.g. basal species).

I would therefore change the title of the paper into - “Efficient conversion of zebrafish

blastomeres into primordial germ cells by overexpression of germplasm components". See next point for why not using the "nine factors" in the title.

Response 2: Yes, thank you for your advice. 'nine germplasm factor' does not fit well in the title. We have changed to "Induced formation of primordial germ cells from zebrafish blastomeres by germplasm factors".

We agree that "Induction" primarily refers to the effect of externally-produced/supplied factors on cells that become germ cells. In fact, the embryonic cells become germ cells by externally-supplied factors "9GMs" in this study, which perfectly reflects the essence of "induction" of PGCs from embryonic cells. Although "preformation" is the mechanism of PGC specification in fish, we show here that a combination of "preformation" factors has PGC induction activity for the non-PGC embryonic cells. Therefore, we believe that "Induction" or "Induced formation" in the revised title can more faithfully express the events of cell fate conversion from embryonic cells to PGCs.

2. Since in their response the authors show that not all 9 factors are needed, it is a pity that the term "9GM" is used, since it will become irrelevant in the very near future and will disappear. It is especially not good to have it in the title. When introducing the converting RNAs I would define the mixture as something like "PGC-converting mix – PCM" and state that it is likely that less than 9 factors are needed and that it is a point for future investigation. This is mentioned at the end of the discussion, but given that a result is already there, better to mention it briefly earlier.

Response 3: Thank you for the suggestion. However, to keep the current manuscript as attractive as it can, we prefer to keep the term "9GMs" in the manuscript. In fact, when we originally prepared the manuscript, we considered that all these 9GMs were necessary for PGC induction, indicating that the term "9GMs" was scientifically reasonable in the current paper. On the other hand, if we define the current mixture as "PGC-converting mix – PCM", it might be difficult for us to clearly state the difference between the future PCM with the current PCM.

3. For the same reason as for the title (point 1), I would not use the term "induced PGCs (iPGCs)" throughout the text, but rather something like "PGCs produced by PCM injection".

Response 4: Thank you for your suggestion. However, we prefer to keep the term "iPGCs". The main reasons have been explained in Response 2.

4. "To our knowledge, this is the first success in inducing PGC in vivo using preformation theory, which creates a new paradigm for PGC induction and PGCT in teleost fish." Change to " Together, we present a novel and efficient method for generating PGCs in a teleost, manipulating them and reintroducing them into host embryos, a technique that will have a strong impact in basic research and aquaculture".

Response 5: Thank you for your suggestion. We have corrected it.

5. "...designed to increase the efficiency of genetic manipulation..". Add an "s" after "manipulation". In general, the paper would very very strongly benefit from English editing, beyond the few examples I mention.

Response 6: Thank you. We have corrected it.

6. References – At many places the authors do not cite the correct source, a point that should be thoroughly addressed. For example – " Maternally inherited germline factors, such as vasa 15, dazl 16, piwil1 17, dnd 18, nanos3 19, tdrd 20, 21, etc., are deposited in presumptive primordial germ cells (pPGCs), leading them to segregate from the somatic lineage and eventually form PGCs.". First, "etc." is used several times instead of "for example" or "such as". In the sentence above, it is not clear that the authors refer to fish only (as the references reveal). Most of the factors mentioned were first found in other organisms. If the statement is general, the corresponding papers should be presented and those in which the molecule was first described. If the authors want to refer only to zebrafish, they should state that and again, cite the first time it was identified as a germ-cell marker in fish. e.g for vasa (a paper from 2014 is cited twice – 15 and 31), instead of DOI: 10.1242/dev.124.16.3157 (a 1997 manuscript). If needed, when referring to actual molecular / cellular function, more recent papers can be cited as well. In contrast, generally, review articles should also be the most recent ones. Overall, the choice of papers to cite is very poor. e.g. reference 25 is not the best one for showing that zebrafish is an important model.

Response 7: Thank you. We have corrected all these references according to your suggestion.

7. "While the induction of PGC-like cells from embryonic stem cells or pluripotent stem cells has been achieved in mice 22, 23, rats 6 and northern white rhinoceros 24, and the induced PGCs (iPGCs) could be further differentiated into functional gametes after transplantation into germ cell-deficient host animals 6, 22, 23, 24." - "While" should not be used at the beginning.

Response 8: Thank you. We have corrected it.

8. "Nevertheless, these induction strategies are all based on the epigenesis model, and there are no successful reports of PGC induction using the preformation model in oviparous species." As mentioned above, better not to use the "induction" in the "preformation" context. In any case, increasing the number of germ cells was described before for example in zebrafish (Buc) and Drosophila (DOI: 10.1038/358387a0 , (where "induction" is used, before the mouse mechanisms were analysed thoroughly)).

Response 9: Thank you. We prefer to keep the term "PGC induction" here. The main reasons have been described in Response 2.

9. In Fig1a – do not use “etc”.

Response 10: Thank you. We have corrected it.

10. “In zebrafish, *Drosophila* and other animals, PGCs are specialized at different sites in the embryo and must migrate to the site of gonadal development for proliferation and differentiation 36, 38.” The PGCs are “specified”, not “specialized”. What does “different sites” mean?

Response 11: Thank you. We have corrected it and deleted the phrase “different sites”.

11. *Cxcr4a* is mentioned as one of the 4 proteins important for migration. This is not the case and it is only *Cxcr4b* (which is not cited - DOI: 10.1016/s0092-8674(02)01135-2)

Response 12: Thank you. We have added the reference for *Cxcr4a* (ref 34).

12. “...further confirming that the germ cells in the host embryos were from the transplanted iPGCs”. Use “were derived” instead of “were”.

Response 13: Thank you. We have corrected it.

13. In Fig 3a, what is the explanation for the fact that control and *buc*-injected cells do not reach the gonad? If the 9 factors (without the “migration proteins”) allow cells to get to the target, what do the control transplanted cells miss? Could state that the 9-factor mix most likely lead to expression of the motility module.

Response 14: Thank you for raising the question. In fact, the control and *buc*-injected cells located in the animal pole, which endogenously lack the PGC mobility module. In contrast, the 9GM-injected cells were converted to PGCs, strongly suggesting that 9GMs overexpression led to expression of the mobility module. We have briefly discussed this point in the revision.

14. In supplemental Fig3, the F2 phenotypes are presumably a result of incrosses – should be indicated in the legend. It is not clear how one obtains the phenotype in F1 (panel d). Is the BMP mutation dominant? please explain, and check if references 42 and 43 are indeed relevant here.

Response 15: Thank you for pointing out this issue. We have added detail explanation in the legend. The F1 generations were obtained by crossing chimera with wild type. The F2 embryos derived from F1 generations through incrossing showed obvious developmental defects (Revised manuscript Supplementary Fig. 3d, e, f).

15. “When Gr_iPGCs were transplanted into ePGC-depleted zebrafish embryos (Gr_iPGCT_Dr), these Gr_iPGCs were able to migrate correctly to the zebrafish genital ridges with a rate of almost 100%, which was much higher than the rate of traditional PGCT (Gr_PGCT_Dr) (Fig. 4c-e and Supplementary Movie 6).”- what one understands from this sentence is that the migration of the Gr_iPGC is more CORRECT than that of “traditional PGCT”. This is not shown in the figures/movie. Rather, the differences in colonization of the gonad are likely to reflect the effect of the larger initial number of cells.

Response 16: Sorry for the lack of clarity in the sentence. We have changed the sentence of “which was much higher than the rate of traditional PGCT (Gr_PGCT_Dr)” to “The percentage of embryos with PGCs in genital ridge using iPGCT was much higher than that using traditional PGCT” (P11, L218-220).

16. “This suggests that the developmental process of germ cells may be faster in Gr_iPGCT_Dr than in Gr 46.” Could this reflect differences in temperature, rather than “faster developmental process”? How does the finding the “the other type expressed Vasa as strongly as Gr, and more than 10 germ cells were grouped together (Fig. 4g)” suggests faster development?

Response 17: Thank you for pointing out this issue. Temperature is indeed one of the important factors affecting the rate of gonad development. We have deleted this sentence in the revision.

17. “It may be due to the fact that germplasm factors derived from zebrafish are used, leading to insufficient.... “ Do the authors mean “used up”?

Response 18: Thank you! We have changed this sentence to “because germplasm factors derived from zebrafish were utilized”.

18. “...assume that the in vivo induction of iPGCs using....” – in vivo should be in italics.

Response 19: Thank you. We have corrected it.

19. The “iPGCT greatly improved the knock-in efficiency of gametes” section fits better before the “Xenogametes generated by iPGCT” part.

Response 20: Thank you for the suggestion! We prefer to keep the “Xenogametes generated by iPGCT” section before the “iPGCT greatly improved the knock-in efficiency of gametes” section, since we did not tried genome editing or gene knock-in in xenograft experiment. We just want to briefly show that the iPGCT approach could be expanded to another fish species.

20. “In zebrafish and medaka fish, previous studies have focused on PGC induction via

single germplasm factors and failed in PGC induction from the essence of PGC formation 33, 59". The use of the word "essence" is not clear here. In addition, Buc over expression was shown to lead to an increase in germ-cell number, albeit not as efficiently as presented in this manuscript.

Response 21: Thank you for pointing out this issue. We have changed the word "essence" to "mechanism".

21. "Midas touch" – better to use a more scientific term.

Response 22: Thank you! We have deleted this phrase here.

Reviewer #3 (Remarks to the Author):

The revised manuscript by Wang et al. has been improved to a satisfactory degree. Now I recommend this study for publication in Nature Communications.

(Minor point)

Figure 3h needs to explain what #1-#5 are in the legend.

Response 23: Thank you for pointing out this issue. #1-#5 refers to the mutation efficiency of 5 F0 fish (#1-#5) gametes. We have added an explanation to the legend (P36-37, L727-729).